# Parvalbumin basket cell myelination accumulates axonal mitochondria to internodes

Koen Kole [1] ✉, Bas J. B. Voesenek [1], Maria E. Brinia[1,2], Naomi Petersen[1] & Maarten H. P. Kole [1,3] ✉

Parvalbumin-expressing (PV+) basket cells are fast-spiking inhibitory interneurons that exert critical control over local circuit activity and oscillations. PV+ axons are often myelinated, but the electrical and metabolic roles of interneuron myelination remain poorly understood. Here, we developed viral constructs allowing cell type-specific investigation of mitochondria with genetically encoded fluorescent probes. Single-cell reconstructions revealed that mitochondria selectively cluster to myelinated segments of PV+ basket cells, confirmed by analyses of a high-resolution electron microscopy dataset. In contrast to the increased mitochondrial densities in excitatory axons cuprizone-induced demyelination abolished mitochondrial clustering in PV+ axons. Furthermore, with genetic deletion of myelin basic protein the mitochondrial clustering was still observed at internodes wrapped by non-compacted myelin, indicating that compaction is dispensable. Finally, two-photon imaging of action potential-evoked calcium (Ca2+) responses showed that interneuron myelination attenuates both the cytosolic and mitochondrial Ca2+ transients. These findings suggest that oligodendrocyte ensheathment of PV+ axons assembles mitochondria to branch selectively fine-tune metabolic demands.

In the vertebrate central nervous system, oligodendrocytes wrap multilamellar membranes around axons to form myelin sheaths, which reduce local internodal capacitance, speed up action potential propagation and enhance action potential fidelity[1,2]. Although the myelin sheath strongly facilitates the electrical conductivity of internodes, the insulation also prevents the axolemma from taking up glucose and other metabolites from the extracellular space. Glycolytic oligodendrocytes aid neurons in overcoming this hurdle by providing trophic support to the axon via the myelin sheath in the form of pyruvate and lactate[3,4], and by modulating the function of axonal mitochondria via exosomes[5]. Mitochondria are versatile organelles that are chiefly committed to the production of adenosine triphosphate (ATP), and in axons require metabolic support from the myelin sheath in order to preserve their integrity[6]. Previous research, mostly performed in white matter tracts and glutamatergic axons, has shown that demyelination

causes dysfunction and shape changes of axonal mitochondria in both multiple sclerosis (MS)[7-9] and experimental models[10-15]. For example, the mitochondrial size and density increase together with electron transport chain protein expression, and mitochondria lose their membrane potential which is critical for ATP synthesis[7,8,11,14]. The accumulation of mitochondria upon demyelination, together with a low mitochondrial content in myelinated axons and vice versa[16-19], have given rise to the general notion that myelination reduces the need for mitochondria.

Myelin is also present around inhibitory gamma-aminobutyric acid (GABA)-ergic axons, and in the neocortex is restricted mostly to parvalbumin-expressing (PV+) basket cells[20-22]. Comprising the most abundant interneuron cell type in the cortex, PV+ basket cells are characterized by a highly complex and extensive axonal organization enabling critical control of the local circuitry and over gamma

---

[1]Axonal Signaling Group, Netherlands Institute for Neuroscience, Royal Netherlands Academy of Arts and Sciences, Meibergdreef 47, 1105 BA Amsterdam, the Netherlands. [2]Medical School, National Kapodistrian University of Athens, Athens 11527, Greece. [3]Cell Biology, Neurobiology and Biophysics, Department of Biology, Faculty of Science, Utrecht University, Padualaan 8, 3584 CH Utrecht, the Netherlands. ✉e-mail: k.kole@nin.knaw.nl; m.kole@nin.knaw.nl

oscillations[23,24]. These interneurons possess small diameter axons (~0.4 μm, ref. 25) which could make them highly vulnerable to pathology of the myelin sheath[26,27]. We recently found that experimental demyelination through toxic or genetic means critically disrupts inhibitory transmission by PV[+] interneurons, characterized by fewer presynaptic terminals, reduced intrinsic excitability and an abolishment of gamma oscillations[22,28]. Interestingly, postmortem studies show that MS is associated with a specific loss in PV[+] interneurons and their presynaptic terminals[28–30]. With a low threshold of action potential generation, sustained high firing frequencies and synchronization with fast gamma oscillations, PV[+] interneurons use large amounts of energy to fuel the sodium/potassium ATPase to maintain the balance of ionic gradients[31,32]. In line with their high energy usage, PV[+] interneurons have a relatively high mitochondrial content and cytochrome c oxidase levels[33,34] while deficits in the spatial distribution of mitochondria in PV[+] axons are associated with impairments of gamma oscillations[35]. One leading hypothesis is that myelination provides trophic support to mitochondria in PV[+] axons to maintain their levels of high-frequency spike generation and activity during gamma oscillations[21,36,37].

To test how myelination affects mitochondrial distribution and function selectively in PV[+] interneurons we developed a Cre-dependent viral approach to express mitochondria-targeted genetically encoded fluorescent reporters, mt-GFP and mt-GCamp6f in Cre-driver lines. Two-photon guided patch-clamp recordings from PV[+] interneurons of the somatosensory cortex followed by confocal microscopy and reconstructions with submicron precision revealed that in normally myelinated axons mitochondria are clustered to myelinated segments, which was disrupted following experimental demyelination. Analyses of a publicly available electron microscopy (EM) dataset[38] confirmed the higher mitochondrial content in myelinated segments of (PV[+]) basket cells but showed homogeneous distribution along partially myelinated layer 2/3 pyramidal neurons. In addition, two-photon imaging of action potential-evoked mitochondrial Ca[2+] transients (mt-GCamp6f) showed that mitochondria in myelinated internodes displayed low or no Ca[2+] buffering activity, which increased following demyelination. These results indicate that in neocortical PV[+] basket cells axons myelination spatially organizes mitochondria, locally increasing the mitochondrial content within internodes, providing evidence for a direct metabolic support by myelination.

## Results

### An AAV-mediated approach for cell type-specific labelling of mitochondria

We first developed a Cre-dependent adeno-associated viral (AAV) vector (AAV-EF1a-mt-GFP-DIO; Fig. 1a) and examined the in vivo labelling of mitochondria. The use of an AAV vector allows control over transduction rates to enable single-cell analyses, an important advantage over brain-wide cell type-specific expression[39]. Upon injection into the somatosensory cortex (S1) of either PV-Cre (labelling PV[+] interneurons) or Rbp4-Cre (labelling cortical layer 5 pyramidal neurons) transgenic mice (Fig. 1b), this viral vector resulted in GFP-labelled mitochondria in molecularly defined cell types (Fig. 1c−f). Expression of mt-GFP-DIO in PV-Cre; Ai14 mice (expressing tdTomato in PV[+] interneurons) labelled interneurons across cortical layers 2 to 6 at high specificity (243/248 cells or 97.94% mt-GFP[+] cells were tdTomato[+]; Fig. 1c; ref. 40). Furthermore, two-photon-guided targeted whole-cell recordings from mt-GFP-expressing (mt-GFP[+]) cells revealed that all cells fired action potentials at high frequency (123.41 ± 13.13 Hz upon 500 pA current injection; n = 8 cells from 5 mice). Action potentials were brief in half-width (295.74 ± 2.52 μs; n = 5 cells from 4 mice) and showed a prominent afterhyperpolarization (−15.83 ± 2.25 mV; n = 5 cells from 4 mice, Fig. 1d), consistent with previously described properties of neocortical PV[+] interneurons[32].

To further test the Cre dependency of the mt-GFP-DIO AAV vector, we expressed mt-GFP[+] in Rbp4-Cre mice, a Cre-driver line commonly used to label cortical layer (L)5 pyramidal neurons (PNs)[41]. In Rbp4-Cre mice, mt-GFP[+] somata were pyramidally shaped, restricted to L5 and possessed a large apical dendrite characteristic of L5 PNs (Fig. 1e; ref. 42). Mt-GFP[+] cells in Rbp4-Cre mice always discharged action potentials in low-frequency trains (40.45 ± 4.76 Hz upon 500 pA current injection; n = 13 cells from 4 mice) and of comparatively long half-width (680.85 ± 15.80 μs; n = 11 cells from 4 mice), consistent with the known electrophysiological properties of L5 PN (Fig. 1d, f; ref. 43). Mitochondria are motile organelles but are substantially more stationary in adult mice and in vivo[44]. We quantified mitochondrial motility in a subset of our mt-GFP[+] acutely prepared slices and found that the vast majority (~99%) of mitochondria were stable in both cell types (Supplementary Fig. 1). Together, these findings indicate that mt-GFP-DIO enables the study of mitochondria in a highly cell-type specific manner.

### Mitochondria are clustered at the axonal initial segment (AIS) and are enlarged upon demyelination

To examine the contribution of myelination to mitochondrial distribution we injected PV-Cre; Ai4 mice with AAV-EF1a-mt-GFP-DIO in S1 and fed mice either with control food or a diet supplemented with 0.2% cuprizone, which selectively kills oligodendrocytes, leading to widespread cortical demyelination[45]. We first focused on the axon initial segment (AIS), a highly excitable domain at the base of the axon where action potentials (APs) are initiated and which is known to adapt to demyelination[43,46]. The AIS was identified by immunostaining for the anchoring protein βIV spectrin in cortical tissue from mt-GFP-expressing PV-Cre; Ai14 mice. By contouring of the perimeter of mt-GFP signals we subsequently estimated the shape and size of mitochondria. The results showed that the mitochondrial area was significantly increased in axons from cuprizone-treated mice (nested t-test, P = 0.0241; Fig. 2a–c). In contrast, the mitochondrial aspect ratio (length/width) and density were unaffected (Fig. 2d, e).

Both the AIS length and distance relative to the soma were comparable between groups (Fig. 2f, g). AIS length and location are critical to the current threshold for AP generation[47]. To test whether AP initiation was preserved we performed whole-cell recordings from PV[+] interneurons in acute slices. In all slice experiments, we recorded features of excitability in artificial cerebral spinal fluid containing 5 mM L-lactate and 10 mM D-glucose, resembling the in vivo extracellular environment containing lactate[48]. In accordance with the unaffected geometrical properties of the AIS, electrophysiological recordings of PV[+] interneurons revealed no change in AP wave form, voltage or current threshold, amplitude, half-width, or resting membrane potential (Fig. 2h, i; Supplementary Fig. 2). However, demyelinated PV[+] interneurons displayed a higher rheobase (Fig. 2j), a lower input resistance (Fig. 2k), and lower firing rate upon sustained somatic current injection (Fig. 2l, m) indicating reduced excitability, in line with previous recordings[22]. Together, these results show that following demyelination PV[+] interneuron excitability is reduced, and AIS mitochondria are larger.

### Mitochondria of the demyelinated PV[+] axon are increased in size and lost in proximal branches

To examine the mitochondrial morphology and distribution throughout the entire cytoarchitecture we reconstructed mt-GFP[+] axons and dendrites of whole-cell recorded biocytin-filled PV[+] interneurons (Supplementary Figs. 3, 4). Axonal mitochondria in PV[+] interneurons were often found at branch points and en passant boutons and were smaller in appearance compared to their dendritic counterparts (Supplementary Fig. 3; ref. 49). Quantification showed that compared to dendritic mitochondria those in axons covered significantly less length, were smaller and more spherical (Supplementary

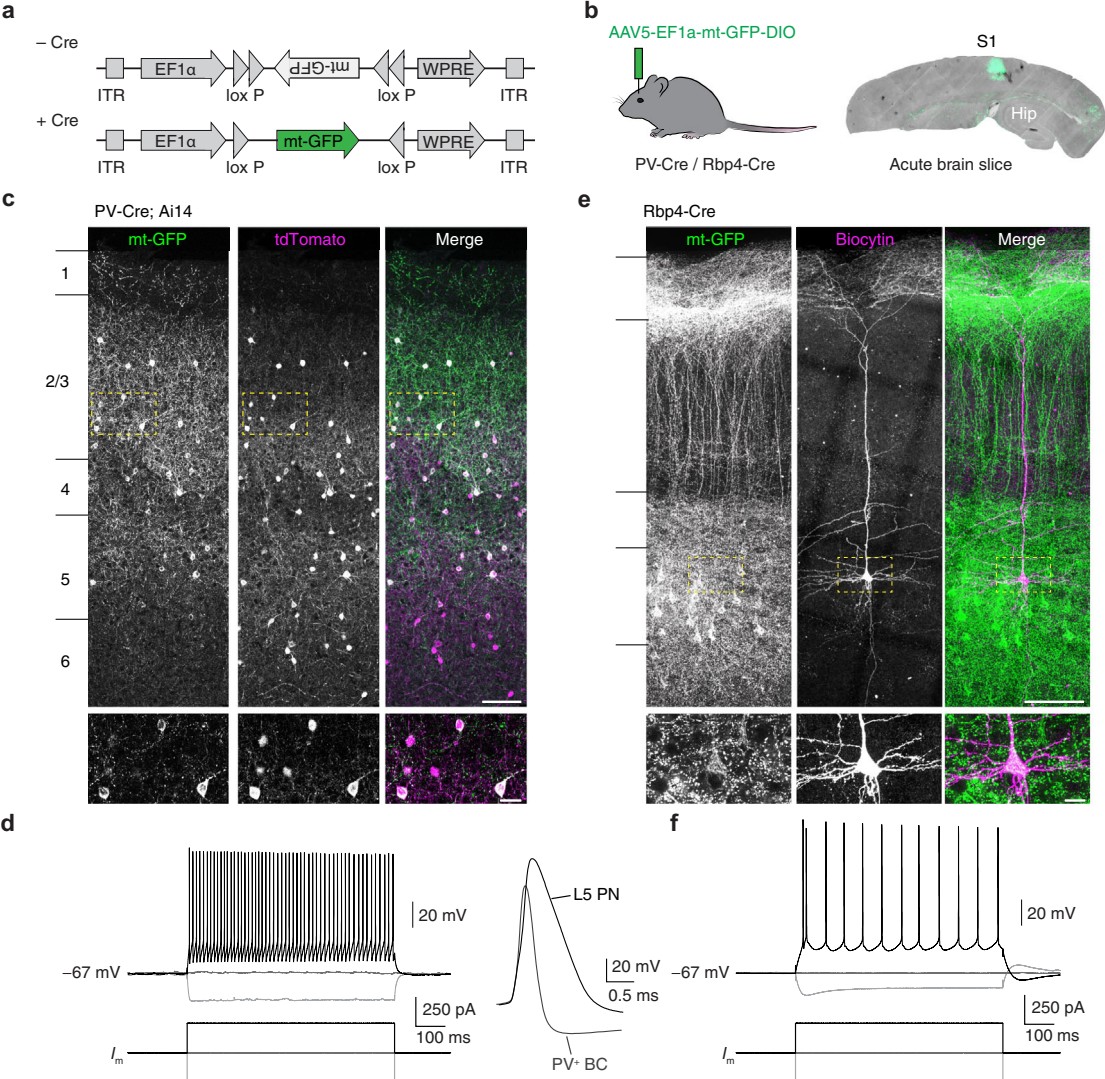

**Fig. 1 | mt-GFP expresses in a Cre-dependent manner. a** Schematic of the AAV construct design. Mitochondria-targeted green fluorescent protein (mt-GFP) requires Cre recombination for expression under the elongation factor-1 alpha (EF1α) promoter. **b** PV-Cre; Ai14 or Rbp4-Cre mice were injected with AAV5-EF1a-mt-GFP-DIO. Hip, hippocampus. **c** Expression in PV-Cre; Ai14 mice results in high overlap between tdTomato and mt-GFP. Scale bar indicates 100 µm. Dashed boxes indicate area depicted by insets at bottom, scale bar indicates 5 µm. Similar results were obtained in >10 mice. **d** Left: Example traces of action potentials in a mt-GFP+ fast spiking PV+ basket cell (BC); right: single action potentials in mt-GFP+ cells from a PV-Cre; Ai14 or Rbp4-Cre mouse (aligned at onset). **e** AAV expression in Rbp4-Cre mice results in mitochondrial labelling in L5 PNs. Note the characteristic apical dendrites visualized by mt-GFP. Scale bar indicates 100 µm. Dashed boxes indicate area depicted by insets at bottom, scale bar indicates 5 µm. Similar results were obtained in >10 mice. **f** Example traces of a current-clamp recording and action potential train in a mt-GFP+ L5 PN.

Fig. 3). These subcellular morphological differences are in agreement with previous findings in L2/3 PNs[50] and the observed mitochondrial properties in L5 PNs (Supplementary Fig. 4).

Consistent with our observations at the AIS, demyelination significantly increased the size of axonal mitochondria (Fig. 3a, b), with a mean surface area increase of ~60% (nested t-test, $P = 0.044$; Fig. 3b, c), without affecting mitochondrial shape (Fig. 3d), suggesting a uniform increase. Importantly, PV+ interneuron myelination is most abundant at lower axonal branch orders (≤4th order)[20,21,25]. To test whether demyelination affects axonal mitochondria differently we separated the distal population from the 2nd to 4th branch orders and found that the demyelination-induced mitochondrial size increase was comparable (Fig. 3f).

The PV axons did not show signs of swellings or pathology and consistent with previous work[22] the total reconstructed axonal length was similar between control and demyelinated axons (unpaired t-test, $P = 0.4918$; Fig. 3g). Interestingly, however, quantification by branch order revealed an increased average segment length in proximal branches of demyelinated PV+ axons (Fig. 3h). The observation of local structural changes, together with previous work showing that demyelination increases mitochondrial density[7,8,12], prompted us to investigate the mitochondrial density in PV+ axons. While the total mitochondrial density across all branches was comparable between control and demyelinated PV+ axons (unpaired t-test, $P = 0.1170$; Fig. 3i), mitochondria in the proximal branch orders of demyelinated axons were on average ~20% reduced in density (two-way ANOVA, branch order × treatment effect, $P = 0.0169$; Fig. 3j). In higher branch orders (≥5) the mitochondrial densities were similar between groups (two-way ANOVA with Bonferroni's post hoc test, $P > 0.9999$). In line with these observations, we found an

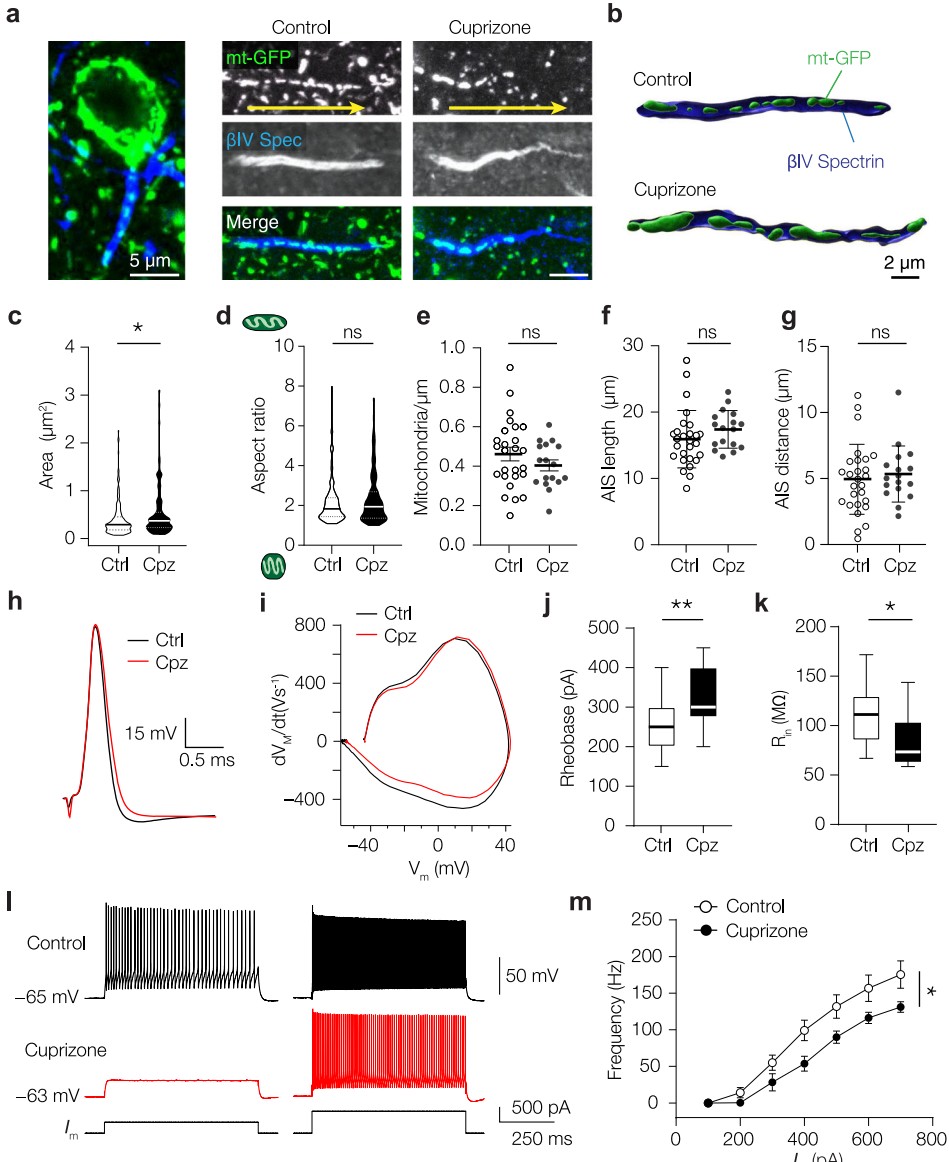

**Fig. 2 | Cuprizone-induced demyelination enlarges AIS mitochondria and reduces PV⁺ interneuron excitability. a** Example images of mitochondria (green) at the AIS (blue) of PV⁺ interneurons. Arrows (yellow) indicate anterograde direction of the axon. Similar results were obtained in 7 mice. **b** 3D surface rendering of PV⁺ AISs containing mitochondria of control or cuprizone-treated axons. **c** Mitochondrial area is increased upon demyelination (two-sided nested t-test, *P = 0.0241; df = 43; Ctrl, n = 215 mitochondria in 27 AISs from 4 mice; Cpz, n = 144 mitochondria in 18 AISs from 3 mice). **d** Mitochondrial aspect ratio is unaffected (two-sided nested t-test, P = 0.5656; df = 43; Ctrl, n = 215 mitochondria in 27 AISs from 4 mice; Cpz, n = 144 mitochondria in 18 AISs from 3 mice). **e** Density of AIS mitochondria is unchanged upon demyelination (two-sided unpaired t-test, P = 0.2396; df = 41; Ctrl, n = 26 AISs from 4 mice; Cpz, n = 17 AISs from 3 mice). **f** AIS length is unaffected by cuprizone treatment (two-sided unpaired t-test, P = 0.2112; df = 42; Ctrl, n = 27 AISs from 4 mice; Cpz, n = 17 AISs from 3 mice). **g** AIS distance (soma edge to AIS) is comparable between control and cuprizone-treated groups (two-sided unpaired t-test, P = 0.6055; df = 42; Ctrl, n = 27 AISs from 4 mice; n = 17 AISs from 3 mice). **h** Example single APs of a control and demyelinated PV⁺

interneuron upon a 3 ms current injection. **i** Phase-plane plots of the traces shown in (**d**); Reduced PV⁺ interneuron excitability suggested by (**j**) increased rheobase (two-sided unpaired t-test, **P = 0.0034; df = 33; Ctrl, n = 22 cells from 12 mice; Cpz, n = 13 cells from 7 mice) and **k** reduced input resistance (two-sided unpaired t-test, *P = 0.0154; df = 33; Ctrl, n = 22 cells from 12 mice; Cpz, n = 13 cells from 7 mice). **l** Example traces of PV⁺ interneuron responses after control (black) or cuprizone (red) treatment in response to 400 (left) or 600 pA (right) current injection. **m** Population data showing a reduced firing frequency upon somatic current injection in demyelinated PV⁺ interneurons (two-way ANOVA with Bonferroni's post hoc test; interaction effect, P = 0.0102, F_{(6, 156)} = 2.910; treatment effect *P = 0.0261, F_{(1, 26)} = 5.569; Ctrl, n = 17 cells from 12 mice; Cpz, n = 11 cells from 7 mice). In truncated violin plots (**c**, **d**), solid lines represent the median, dotted lines represent 25th and 75th quartiles. Scatter plots (**e**–**g**) indicate the mean, individual data points represent cells, error bars indicate SEM. Line graphs (**m**) indicate means, error bars indicate SEM. Box plots (**j**, **k**) represent the 25th to 75th percentiles, whiskers indicate the maximal and minimal values, solid line represents the median. Source data are provided as a Source data file.

increased inter-mitochondrial distance specifically in proximal branches of demyelinated axons (≤4th order; Kruskal–Wallis test with Dunn's post hoc test, P < 0.0001; Fig. 3k). These results are unexpected since previous studies consistently reported an increase of mitochondria in demyelinated axons[12,13]. To examine mitochondrial changes specifically in glutamatergic L5 axons we used a similar

viral strategy in Rbp4-Cre mice and examined the white matter below the somatosensory cortex. In contrast to the mitochondrial loss in PV⁺ interneurons we found a higher density of mitochondria after cuprizone-mediated demyelination (unpaired t-test, P = 0.0441; Supplementary Fig. 5). Together, these results indicate that myelin loss results in cell-type distinct changes, reducing mitochondrial

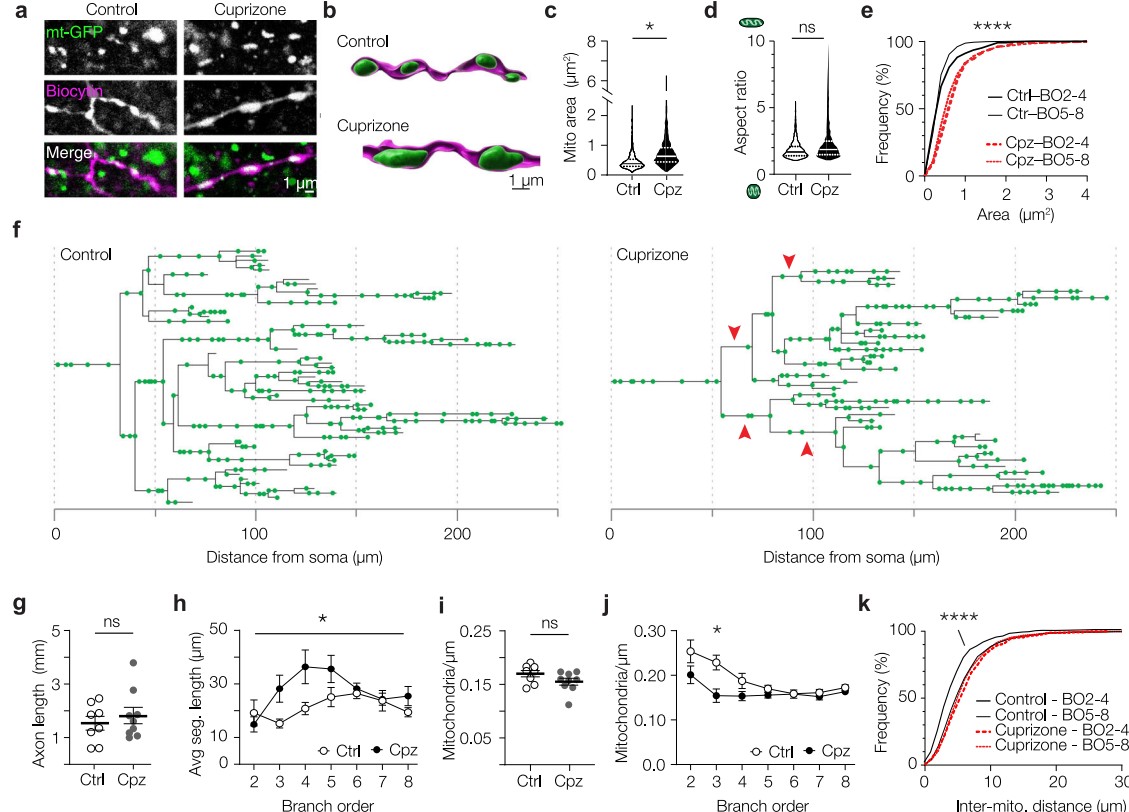

**Fig. 3 | Cuprizone-induced demyelination reduces mitochondrial density in proximal PV⁺ interneuron axons. a** Example confocal images of mitochondria (mt-GFP) in control (left) and demyelinated axon (right). Similar results were obtained in 7 mice. **b** 3D rendered images of mt-GFP and biocytin. **c** Increased mitochondrial size upon demyelination (two-sided nested t-test, *$P = 0.0440$; df = 8; Ctrl, $n = 5$ axons, 805 mitochondria; 5 cells from 3 mice, cuprizone (Cpz), $n = 5$ axons, 1218 mitochondria; 5 cells from 4 mice). **d** Unchanged mitochondrial aspect ratio in demyelinated axons (two-sided nested t-test, $P = 0.2962$; df = 8; Ctrl, $n = 5$ axons, 805 mitochondria; 5 cells from 3 mice; Cpz, $n = 5$ axons, 1218 mitochondria; 5 cells from 4 mice). **e** Cumulative distribution plots of mitochondrial size showing uniform increase across all branch orders (BO, Kruskal–Wallis test ****$P < 0.0001$; Dunn's post hoc test, All Ctrl vs Cpz comparisons, $P < 0.0001$, Ctrl BO 2-4 vs BO 5-8, $P = 0.8402$; Cpz BO 2-4 vs BO 5-8, $P = 0.3932$. Ctrl, $n = 5$ axons, 805 mitochondria; 5 cells from 3 mice. Cpz, $n = 5$ axons, 1218 mitochondria; 5 cells from 4 mice). **f** Example axonograms of a control- (left) and a cuprizone-treated (right) PV⁺ interneuron. Red arrowheads indicate examples of second, third or fourth branch orders where mitochondria appear relatively sparse. **g** Cuprizone treatment induces no change in axonal length (two-sided unpaired t-test, $P = 0.4918$; df = 15; Ctrl, $n = 8$ axons, 5 mice; Cpz, $n = 9$ axons, 5 mice). **h** Increased segment length in demyelinated PV⁺ axons as a function of branch order (two-way ANOVA; branch

order x treatment effect, *$P = 0.0440$, $F_{(6,90)} = 2.266$; branch order effect, $P = 0.0104$, $F_{(3.916, 58.71)} = 3.660$; treatment effect, $P = 0.0479$, $F_{(1,15)} = 4.639$; Bonferroni's post hoc test n.s.; Ctrl, $n = 8$ axons, 5 mice; Cpz, $n = 9$ axons, 5 mice). **i** No change in overall axonal mitochondrial density (two-sided unpaired t-test, $P = 0.1170$; df = 15; Ctrl, $n = 8$ axons, 5 mice; Cpz, $n = 9$ axons, 5 mice). **j** Reduced mitochondrial density in proximal (<5) but not distal branch orders (two-way ANOVA, Branch order x treatment effect, $P = 0.0169$, $F_{(6,90)} = 2.747$; branch order effect, $P < 0.0001$, $F_{(2.764, 41.45)} = 9.560$; treatment effect, $P = 0.0143$, $F_{(1,15)} = 7.679$; Bonferroni's post hoc test Branch order 3, *$P = 0.0340$; Ctrl, $n = 8$ axons, 5 mice; Cpz, $n = 9$ axons, 5 mice). **k** Cumulative distribution plot showing increased distance between mitochondria in second to fourth branch orders of demyelinated axons but no change in later branch orders (BO, Kruskal–Wallis test $P < 0.0001$; Dunn's post hoc test, Ctrl vs Cpz BO2-4, ****$P < 0.0001$; Ctrl BO2-4 vs Ctrl BO5-8, $P < 0.0001$; Cpz BO2-4 vs Cpz BO5-8, $P = 0.6008$; Ctrl, $n = 8$ axons, 5 mice; Cpz, $n = 9$ axons, 5 mice). Solid lines in truncated violin plots (**c, d**) represent the median, dotted lines represent 25th and 75th quartiles. Line graphs (**h, j**) indicates means, error bars indicate SEM. Horizontal bars (**g, i**) represent the mean, individual data points represent axons, error bars represent SEM. Source data are provided as a Source data file.

clustering specifically in the proximal and putatively demyelinated branches of the PV⁺ axon.

## Mitochondria are clustered at the myelinated PV⁺ axon

The loss of mitochondrial clustering in proximal axons following demyelination suggests that their distribution is determined, at least in part, by the presence of the myelin sheath. This should be visible in the proximal arbours of PV⁺ interneurons, which are characterized by heterogeneous myelin patterns[20,21,25]. To test this, we performed a triple staining for myelin basic protein (MBP), GFP and biocytin in sections from control mice and reconstructed PV⁺ axons to quantify the mitochondrial density in MBP positive (MBP⁺) and negative (MBP⁻) axonal segments (Fig. 4a, b, Fig S6). In agreement with previous findings[20,21], MBP⁺ segments were ~19 μm in length (on average $18.59 \pm 1.84$ μm, range: 2.56–47.62 μm, $n = 32$ internodes). Interestingly,

axonal mitochondria in myelinated segments were larger (nested t-test, $P = 0.0121$; Fig. 4c), and more elongated compared to mitochondria in unmyelinated branches (nested t-test, $P = 0.0388$; Fig. 4d). We observed that MBP⁺ segments contained more mitochondria (paired t-test, $P = 0.0013$, Fig. 4e), and were also significantly longer than MBP⁻ segments (paired t-test, $P = 0.0267$, Fig. 4f). After correcting for length, MBP⁺ axonal segments displayed a significantly higher mitochondrial density compared to MBP⁻ segments of the same axon and comparable branch orders (Fig. 4a, b, g). Furthermore, mitochondrial densities in control MBP⁻ segments were not different from cuprizone-induced demyelinated PV⁺ axons (one-way ANOVA, $P = 0.0138$; Bonferroni's post hoc test, MBP⁻ vs. MBP⁺ $P = 0.0356$, MBP⁺ vs. cuprizone BO3 $P = 0.0276$; MBP⁻ vs cuprizone BO3 $P > 0.9999$). These results support the idea that myelin wrapping alone suffices to cluster and increase the size of mitochondria locally within PV⁺ internodes.

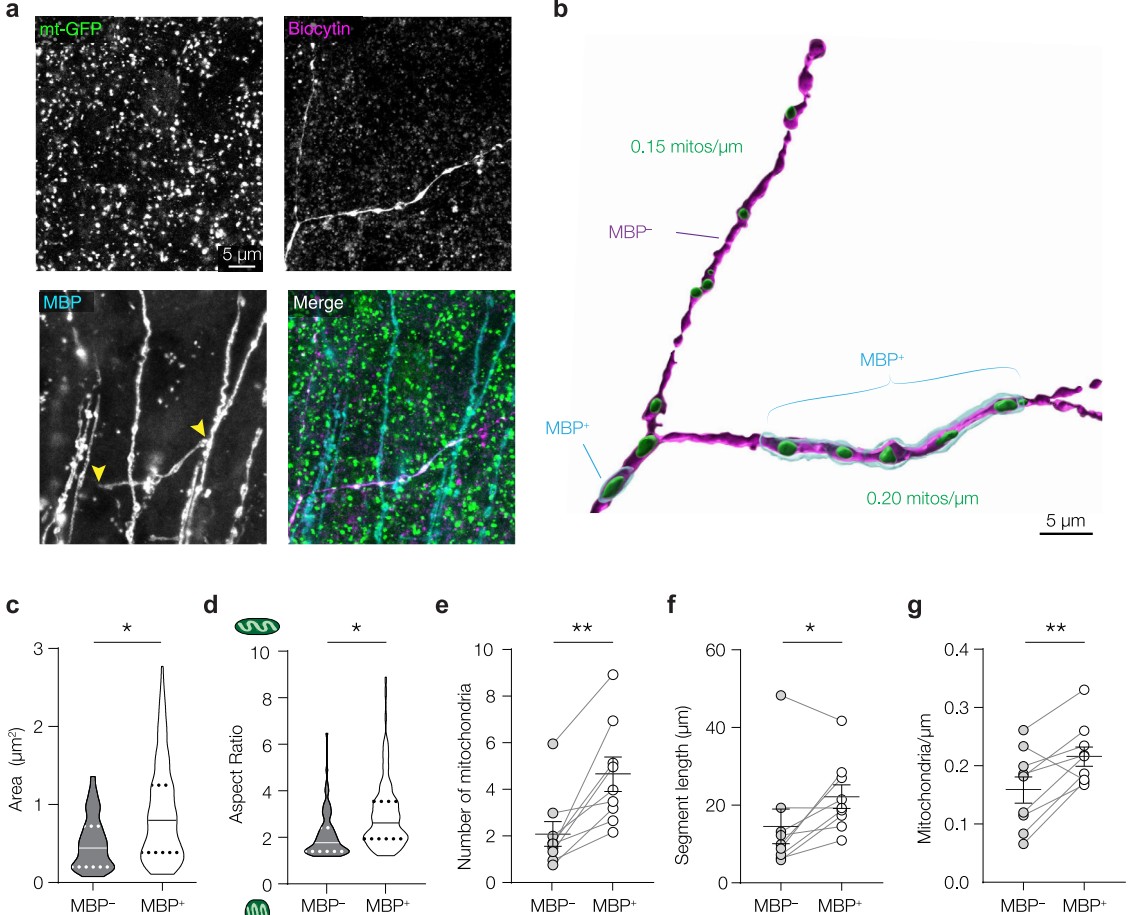

**Fig. 4 | Mitochondria are clustered to myelinated segments of PV⁺ axons.**
**a** Confocal images of a control PV⁺ axon branch point with one myelinated (MBP⁺) and one unmyelinated (MBP⁻) daughter branch. Similar results were obtained in 6 mice. **b** 3D surface rendering of the confocal images in (**a**). **c** Mitochondrial area is higher at the myelin sheath (two-sided nested t-test, $P = 0.0121$; df = 14; unmyelinated, $n = 68$ mitochondria; myelinated, $n = 114$ mitochondria, 8 axons from 6 mice). **d** Mitochondrial aspect ratio is significantly higher in myelinated axons (two-sided nested t-test, $P = 0.0388$; df = 14; unmyelinated, $n = 68$ mitochondria; myelinated, $n = 114$ mitochondria, 8 axons from 6 mice). **e** More mitochondria in MBP⁺ segments (two-sided paired t-test, **$P = 0.0013$, df = 8; $n = 9$ axons from 6 mice). **f** Myelinated PV⁺ axonal segments are significantly longer (two-sided paired t-test, *$P = 0.0267$, df = 8; $n = 9$ axons from 6 mice). **g** Mitochondrial density in the myelinated segment is higher compared to unmyelinated ones within the same axon (two-sided paired t-test, **$P = 0.0099$; $n = 9$ axons from 6 mice). Solid lines in truncated violin plots (**c, d**) represent the median, dotted lines represent 25th and 75th quartiles. Individual data points (**e–g**) represent axons (1–4 segments per axon), horizontal bars indicate means, error bars indicate SEM. Source data are provided as a Source data file.

## Myelination clusters mitochondria selectively to baskets cell internodes

The unexpected axonal mitochondrial clustering within myelinated PV⁺ BC internodes could be a common feature of axon-glia signalling along intermittently myelinated axons. To test this with alternative methods we examined a fully annotated and minable 3D electron microscopy (EM) dataset of the mouse visual cortex[38]. We identified (putative PV⁺) basket cells based on their morphological properties and presence of myelinated arbours (Fig. 5a; see 'Methods') and compared axonal mitochondrial distribution with those of the primary axons of L2/3 PNs which are known to exhibit patchy distribution of myelin along the primary axon[51] (Fig. 5b).

Consistent with our immunofluorescence analysis, basket cell myelinated segments were significantly longer (two-way ANOVA with Bonferroni's post hoc test, $P = 0.0004$, Supplementary Fig. 6). Within basket cell axons mitochondria were found in ~60% of branch points with at least one myelinated branch (presumed nodes; Supplementary Fig. 6). Interestingly, mitochondria typically avoided ~2 μm regions near the edges of the myelin sheath corresponding to the paranodal loops[17,52], but were with high probability and high densities distributed within the internode (Supplementary Fig. 6). Myelinated segments of basket cells, both quantitatively and qualitatively in line with the immunoanalysis (Fig. 4), had an increased mitochondrial density compared to unmyelinated ones (two-way ANOVA with Bonferroni's post hoc test, $P = 0.0130$, Fig. 5c). Interestingly, mitochondrial density in myelinated internodes of L2/3 PNs was not different compared to upstream or downstream segments that were unmyelinated (two-way ANOVA with Bonferroni's post hoc test, $P = 0.1665$, Fig. 5c).

Furthermore, in keeping with our immunofluorescence data (c.f. Fig. 4), mitochondria within myelinated basket cell segments were significantly larger ($P = 0.0087$, Fig. 5d, Supplementary Fig. 6). Mitochondrial size was, however, not affected in L2/3 PN axonal segments (two-way ANOVA with Bonferroni's post hoc test, $P > 0.9999$, Fig. 5d). Strikingly, when comparing the mitochondria between the two cell types, those in basket cells were ~7-fold larger (two-way ANOVA with Bonferroni's post hoc test, $P = 0.0009$, Fig. 5d). Consistent with previous reports[25] we noticed that unmyelinated segments were typically thinner. A lower cytoplasmic volume could reduce the need for mitochondria and explain the lower density and smaller size of mitochondria. However, plotting the ratio of mitochondrial to axonal volume showed that mitochondria in myelinated basket cell axons occupied a significantly larger volume compared to unmyelinated branches (two-way ANOVA with Bonferroni's post hoc test, $P = 0.0044$, Fig. 5e). In contrast, in L2/3 PNs the

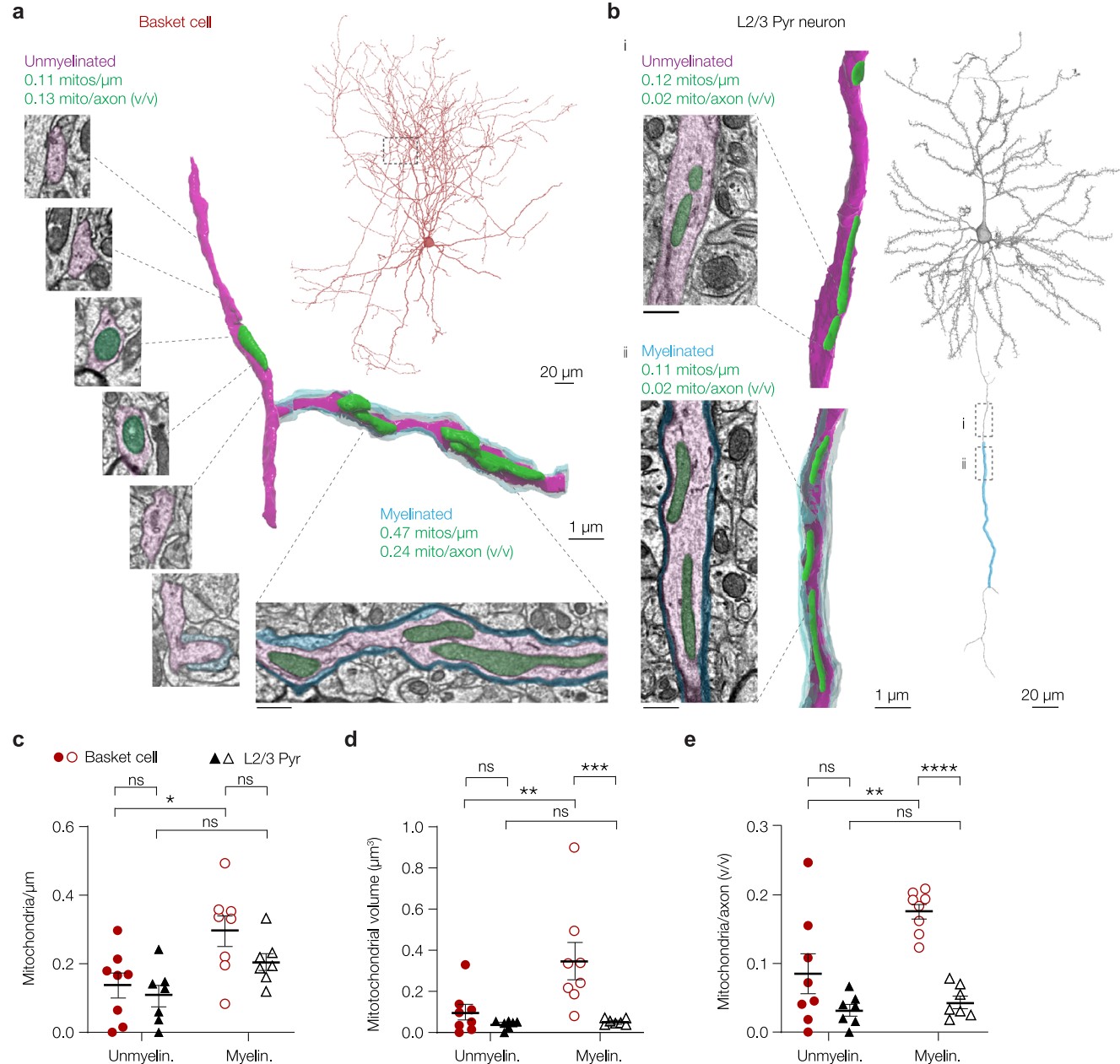

**Fig. 5 | Myelination assembles mitochondria in basket cell- but not PN internodes. a, b** 3D EM renders and reconstructions of a basket cell (**a**) and L2/3 PN (**b**) axon with and without myelin. Similar results were obtained in 8 basket cells and 7 L2/3 PNs. **c** Mitochondrial density is increased in myelinated basket but not L2/3 PN segments (two-way ANOVA, myelination x cell type effect, $P = 0.4162$, $F_{(1,13)} = 0.7052$; myelination effect, $P = 0.0034$, $F_{(1,13)} = 12.82$; cell type effect, $P = 0.1206$, $F_{(1,13)} = 2.760$; Bonferroni's post hoc test, basket myelin. vs. unmyelin., $*P = 0.0130$; PN myelin. vs. unmyelin., $P = 0.1665$; Myelin. basket vs PN, $P = 0.1766$; Unmyelin. basket vs PN $P > 0.9999$). **d** Mitochondria in basket myelinated axonal segments are larger compared to those in PNs (two-way ANOVA, myelination x cell type effect, $P = 0.0430$, $F_{(1,13)} = 5.028$; myelination effect, $P = 0.0285$, $F_{(1,13)} = 6.070$; cell type effect, $P = 0.0043$, $F_{(1,13)} = 11.89$; Bonferroni's post hoc test, basket myelin. vs. unmyelin., $**P = 0.0087$; PN myelin. vs. unmyelin., $P > 0.9999$; Myelin. basket vs PN, $***P = 0.0009$; Unmyelin. basket vs PN $P = 0.9090$; basket, myelin. $n = 68$, unmyelin.

$n = 18$ mitochondria; PN, myelin. $n = 53$, unmyelin. $n = 21$ mitochondria). **e** In myelinated basket but not PN segments mitochondria take up a larger fraction of the axonal volume (two-way ANOVA, myelination x cell type effect, $P = 0.0410$, $F_{(1,13)} = 5.143$; myelination effect, $P = 0.0117$, $F_{(1,13)} = 8.578$; cell type effect, $P = 0.0002$, $F_{(1,13)} = 27.17$; Bonferroni's post hoc test, basket myelin. vs. unmyelin., $**P = 0.0044$; PN myelin. vs. unmyelin., $P > 0.9999$; Myelin. basket vs PN, $****P < 0.0001$; Unmyelin. basket vs PN $P = 0.0837$). Scale bars for EM images in **a** and **b** represent 0.5 μm. Data points in **c** and **e** represent axons (basket, $n = 8$ cells; PN, $n = 7$ cells; 1–4 segments per cell for each cell type), data points in **d** represent average mitochondrial volume per cell (basket, $n = 8$ cells, $n = 68$ mitochondria in myelinated and 18 mitochondria in unmyelinated segments; PN, $n = 7$ cells; $n = 21$ mitochondria in myelinated and 53 in unmyelinated segments). Solid horizontal bars indicate means, error bars indicate SEM. Source data are provided as a Source data file.

relative volume of mitochondria was comparable between myelinated and unmyelinated segments (two-way ANOVA with Bonferroni's post hoc test, $P > 0.9999$, Fig. 5e).

Taken together, the 3D EM results provide independent support of our AAV immunofluorescence results, revealing at ultrastructural

detail that mitochondria selectively cluster to myelinated PV⁺ basket cell internodes. Furthermore, they suggest that the clustering of large mitochondria at myelinated axonal segments is not merely a consequence of patchy myelination but may be specific to PV⁺ basket cells.

## Compact myelin is not required for mitochondrial clustering to myelinated PV+ axons

Our data so far suggest that PV+ oligodendroglia myelination directly controls mitochondrial clustering. Axonal mitochondria are supplied with metabolites from glycolytic oligodendrocytes via a system of cytoplasmic channels within myelin and transported across the adaxonal membrane and axolemma[3,4,53]. To test the role of compact and noncompact myelin we used the *Shiverer* mouse line. These mice harbour a deletion in the *Mbp* gene severely reducing MBP protein levels and resulting in myelin wrapping with only a few noncompacted oligodendroglial membranes[54,55]. We crossed PV-Cre; Ai14 mice with *Shiverer* mice (PV-Cre; Ai14 × Shiverer) and injected wildtype (*Mbp*WT) and homozygous (*Mbp*Shi) mice with AAV-EF1a-mt-GFP-DIO to target GFP to mitochondria of normal or dysmyelinated PV+ interneurons. Patch-clamp recordings from PV+ interneurons revealed no differences between *Mbp*WT or *Mbp*Shi mice for any of the electrophysiological properties (Supplementary Fig. 7). During whole-cell recordings, cells were filled with biocytin and to identify the location of (noncompacted) myelin lamellae we performed a triple staining for biocytin, GFP and myelin oligodendrocyte glycoprotein (MOG), a protein which is reduced but still expressed in *Mbp*Shi mice (Fig. 6a–c; Supplementary Fig. 7; ref. 56).

Mitochondrial distribution in PV+ interneurons from *Mbp*WT mice showed a comparable pattern as found in control mice (Fig. 6 c.f. Fig. 5). In brief, mitochondria were both larger and longer in MOG+ segments (Fig. 6d, e). Mitochondria were present in higher numbers in myelinated segments, and myelinated axons were significantly longer than their unmyelinated counterparts (Fig. 6f, g). Plotting mitochondria per unit length showed that myelinated axons displayed higher mitochondrial densities compared to those lacking myelin (Fig. 6h).

In the *Mbp*Shi group, mitochondrial morphology was also myelin-dependent with larger and more tubular mitochondria in MOG+ segments (two-way ANOVA with Bonferroni's post hoc test; area, $P = 0.0001$, aspect ratio, $P = 0.0069$; Fig. 6d, e). In contrast, the number of mitochondria in MOG− and MOG+ axonal arbours were comparable (two-way ANOVA with Bonferroni's post hoc test, $P = 0.2549$; Fig. 6f). Segments of PV+ axons possessing noncompacted myelin were also shorter in *Mbp*Shi mice (two-way ANOVA with Bonferroni's post hoc test, $P = 0.0315$; Fig. 6g), and there was no difference in length between MOG+ and MOG− segments (two-way ANOVA with Bonferroni's post hoc test, $P > 0.9999$; Fig. 6g). Interestingly, when correcting for segment length, there was no difference in mitochondrial density between noncompact myelinated and unmyelinated axons (two-way ANOVA with Bonferroni's post hoc test, $P = 0.1070$, Fig. 6h), due to a

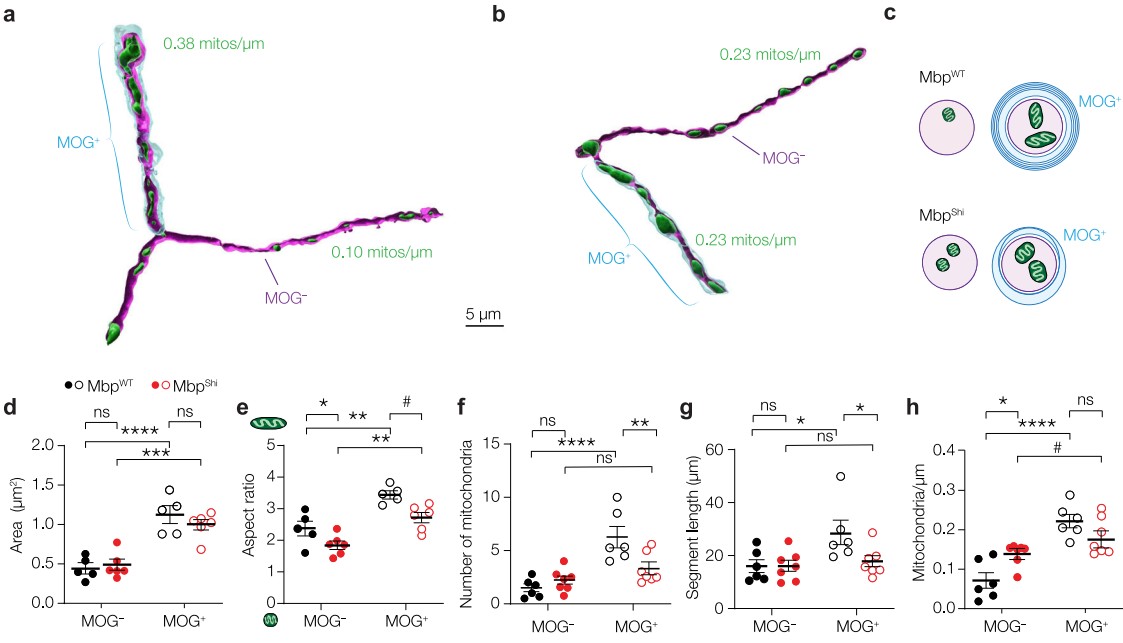

**Fig. 6 | Noncompacted myelin suffices for internodal mitochondrial clustering.**
**a**, **b** Example 3D renders of a PV+ axon of a (**a**) *Mbp*WT or (**b**) *Mbp*Shi mouse (see Supplementary Fig. 7). **c** Schematic representation of mitochondrial distributions in *Mbp*WT and *Mbp*Shi PV+ axons. **d** Mitochondria under myelin sheaths are larger in both *Mbp*WT and *Mbp*Shi mice (two-way ANOVA, myelination x genotype effect, $P = 0.1481$, $F_{(1, 9)} = 2.504$; myelination effect, $P < 0.0001$, $F_{(1, 9)} = 125.5$; genotype effect, $P = 0.6304$, $F_{(1, 9)} = 0.2480$; Bonferroni's post hoc test, *Mbp*WT MOG− vs. MOG+, ****$P < 0.0001$; *Mbp*Shi MOG− vs. MOG+, ***$P < 0.0001$; MOG− *Mbp*WT vs. *Mbp*Shi, $P > 0.9999$; MOG+ *Mbp*WT vs. *Mbp*Shi, $P = 0.4792$; *Mbp*WT, $n = 5$ cells from 3 mice, MOG+, $n = 72$ mitochondria, MOG−, $n = 28$ mitochondria; *Mbp*Shi, $n = 6$ cells from 3 mice, MOG+, $n = 78$ mitochondria, MOG−, $n = 59$ mitochondria). **e** Internode mitochondria in both *Mbp*WT and *Mbp*Shi mice are longer compared to those in unmyelinated axonal segments (two-way ANOVA, myelination x genotype effect, $P = 0.5428$, $F_{(1, 9)} = 0.4000$; myelination effect, $P = 0.0002$, $F_{(1, 9)} = 35.16$; genotype effect, $P = 0.0045$, $F_{(1, 9)} = 14.13$; Bonferroni's post hoc test, *Mbp*WT MOG− vs. MOG+, **$P = 0.0032$; *Mbp*Shi MOG− vs. MOG+, **$P = 0.0069$; MOG− *Mbp*WT vs. *Mbp*Shi, #$P = 0.0825$; MOG+ *Mbp*WT vs. *Mbp*Shi, *$P = 0.0125$; *Mbp*WT, $n = 5$ cells from 3 mice, MOG+, $n = 72$ mitochondria, MOG−, $n = 28$ mitochondria; *Mbp*Shi, $n = 6$ cells from 3 mice, MOG+, $n = 78$ mitochondria, MOG−, $n = 59$ mitochondria). **f** Number of mitochondria in PV+ axons of *Mbp*WT and *Mbp*Shi mice (two-way ANOVA, myelination

x genotype effect, $P = 0.0027$, $F_{(1, 11)} = 14.77$; myelination effect, $P < 0.0001$, $F_{(1, 11)} = 37.00$; genotype effect, $P = 0.1406$, $F_{(1, 11)} = 2.522$; Bonferroni's post hoc test, *Mbp*WT MOG− vs. MOG+, ****$P < 0.0001$; *Mbp*Shi MOG− vs. MOG+, $P = 0.2549$; MOG− *Mbp*WT vs. *Mbp*Shi, $P = 0.7802$; MOG+ *Mbp*WT vs. *Mbp*Shi, **$P = 0.0041$; *Mbp*WT, $n = 6$ cells from 3 mice; *Mbp*Shi, $n = 7$ cells from 3 mice). **g** Segment lengths of PV+ interneuron axons in *Mbp*WT or *Mbp*Shi mice (two-way ANOVA, myelination x genotype effect, $P = 0.0486$, $F_{(1, 11)} = 4.914$; myelination effect, $P = 0.0148$, $F_{(1, 11)} = 8.330$; genotype effect, $P = 0.1348$, $F_{(1, 11)} = 2.605$; Bonferroni's post hoc test, *Mbp*WT MOG− vs. MOG+, *$P = 0.0104$; *Mbp*Shi MOG− vs. MOG+, $P > 0.9999$; MOG− *Mbp*WT vs. *Mbp*Shi, $P > 0.9999$; MOG+ *Mbp*WT vs. *Mbp*Shi, *$P = 0.0315$; *Mbp*WT, $n = 6$ cells from 3 mice; *Mbp*Shi, $n = 7$ cells from 3 mice). **h** Mitochondrial density is higher in internodes in both *Mbp*WT and *Mbp*Shi mice (two-way ANOVA, myelination x genotype effect, $P = 0.0034$, $F_{(1, 11)} = 13.87$; myelination effect, $P < 0.0001$, $F_{(1, 11)} = 44.39$; genotype effect, $P = 0.4762$, $F_{(1, 11)} = 0.5441$; Bonferroni's post hoc test, *Mbp*WT MOG− vs. MOG+, ****$P < 0.0001$; *Mbp*Shi MOG− vs. MOG+, $P = 0.1070$; MOG− *Mbp*WT vs. *Mbp*Shi, *$P = 0.0168$; MOG+ *Mbp*WT vs. *Mbp*Shi, $P = 0.1919$; *Mbp*WT, $n = 6$ cells from 3 mice; *Mbp*Shi, $n = 7$ cells from 3 mice). Solid horizontal bars indicate means, error bars indicate SEM, individual data points indicate cells. Source data are provided as a Source data file.

significantly increased mitochondrial density in the unmyelinated segments (two-way ANOVA with Bonferroni's post hoc test, $P = 0.0168$, Fig. 6h). These results provide further evidence that mitochondria are heterogeneously distributed along the proximal arborization of PV[+] axons as a function of myelination. In addition, the presence of non-compacted membranes suffices for mitochondrial assembly at internodes.

### Myelination attenuates mitochondrial Ca[2+] buffering

The clustering of mitochondria to internodes may be important for the buffering of Ca[2+] during neuronal activity[50]. Our recent work showed that myelin strongly attenuates but not completely abolishes depolarization of the axolemma enabling activation of internode voltage-gated channels near the node of Ranvier[1]. Action potential generation has been shown to cause internodal Ca[2+] influx spreading far from the node of Ranvier[57,58] but to which extent this holds true for PV[+] axons and how mitochondria within and outside internodes respond to activity-dependent Ca[2+] influx is unknown.

To examine mitochondrial buffering of Ca[2+] in PV[+] interneurons we employed AAV-mediated and Cre-dependent expression of the mitochondrion-targeted, genetically encoded calcium indicator mt-GCaMP6f in PV[+] interneurons (Fig. 7). In acute slices of PV-Cre; Ai14 mice we targeted mt-GCaMP6f[+] PV[+] interneurons for whole-cell patch-clamp recordings and filled them with Atto594, visualizing the arborization of axons during 2 P recordings and enabling targeted mitochondrial imaging (Fig. 7a). In the experiments we imaged mt-Ca[2+] fluorescence responses following a train of ~100 APs (~143 Hz). In line with previous findings, we observed that during AP trains mitochondria in putative *en passant* boutons of the distal PV[+] axon showed strong Ca[2+] responses, which were unaffected upon demyelination (Supplementary Fig. 8; refs. 50, 59). Similarly, the large amplitude of mt-Ca[2+] transients at the AIS of PV[+] interneurons, was unchanged after demyelination (Supplementary Fig. 8).

To test whether there are differences between internodes, unmyelinated and/or demyelinated branches we imaged mt-GCaMP6f along the proximal arbors (≤5th branch order) and post hoc immunostained the slices for GFP, MBP and biocytin, allowing unequivocal assessment of the location of mitochondria with respect to myelin sheaths (Fig. 7b–d). We first focused on segments (i.e. excluding branch points) and found that mitochondria in control axons showed AP-evoked Ca[2+] responses with greater amplitudes within MBP[−] segments compared to those in internodes (Kruskal–Wallis test with Dunn's post hoc test, $P < 0.0001$; Fig. 7d, Supplementary Movie 1). We next asked whether myelin loss would lead to changes in mt-Ca[2+] buffering in demyelinated PV[+] interneurons. When we compared all mt-Ca[2+] responses in control axons (including both MBP[+] and MBP[−] segments) with those in putatively demyelinated axons we found no difference between the two groups (nested t-test, $P = 0.6055$). However, mt-Ca[2+] transients in cuprizone-treated axons were on average larger compared to MBP[+] control segments (Kruskal–Wallis test with Dunn's post hoc test, $P = 0.0039$; Fig. 7d) but smaller in amplitude compared to MBP[−] segments (Kruskal–Wallis test with Dunn's post hoc test, $P = 0.0195$; Fig. 7d). Mitochondrial Ca[2+] uptake is known to depend on the influx of extracellular Ca[2+] (refs. 59, 60), which in turn is controlled by myelination[57]. To examine a possible source for the changes in the mt-Ca[2+], we next examined cytosolic Ca[2+] (cyt-Ca[2+]) transients in PV[+] interneurons. To this end, we expressed GCaMP6f in the cytosol of PV[+] interneurons and imaged the axonal AP-evoked Ca[2+] transients in MBP[+], MBP[−] or putatively demyelinated axonal segments. Similar to our results in mitochondria, cyt-Ca[2+] transient amplitudes were weak in MBP[+] arbours (Kruskal–Wallis test with Dunn's post hoc test, $P < 0.0001$, Fig. 7f, Supplementary Movie 2), while demyelinated axons showed increased Ca[2+] responses compared to MBP[+] segments (Kruskal–Wallis test with Dunn's post hoc test, $P = 0.0004$, Fig. 7f). These results suggest that myelination strongly attenuates internodal Ca[2+] entry, limiting downstream mt-Ca[2+] responses in control PV[+] axons.

### Myelin loss impairs mitochondrial Ca[2+] responses at branch points

Upon closer inspection of the localization of the mitochondria with large mt-Ca[2+] responses we found that they were near the outer edge of the myelin sheath or at branch points (Figs. 7b, 8a). To further identify the precise localization of the myelin borders we immunostained for Caspr, a marker for paranodes, which typically had a length of $2.82 \pm 0.18\,\mu m$ (Fig. 8b; $n = 36$ Caspr[+] segments of 3 cells from 2 mice). The Caspr[+] segments flanked branch points if their branches were myelinated ($n = 19$ branch points of 4 cells from 3 mice, Fig. 8b). In line with our initial observation, mitochondria were located near Caspr[+] signals (range $0.00–16.57\,\mu m$, on average $2.46 \pm 0.96\,\mu m$ distance, $n = 20$ paranodes of 4 cells from 3 mice; Fig. 8c). These data suggest that in PV[+] axons branch points are sites of nodes of Ranvier and mitochondria are often found at the nodal domains (Supplementary Fig. 7).

Finally, demyelination is associated with a disassembly of proteins of the node of Ranvier[43,61,62]. In PV[+] axons cuprizone-induced demyelination caused a complete loss of Caspr[+] segments flanking branch points (0 Caspr[+] segments in 4 cells from 3 mice; Fig. 8d). To investigate how the nodal mt-Ca[2+] transients were affected we compared branch point mt-Ca[2+] transients[42,63]. The results showed that the AP-evoked mt-Ca[2+] transients in branch points of demyelinated PV[+] axons were lower in amplitude (Mann–Whitney test, $P = 0.0123$; Fig. 8e, f). Moreover, in putatively demyelinated PV[+] axons, mt-Ca[2+] responses in branch points were indistinguishable from those in demyelinated axonal segments (Mann–Whitney test, $P = 0.8106$, Fig. 8g). Similar to the mt-Ca[2+] responses, we observed that cyt-Ca[2+] transients at branch points were reduced upon demyelination (Mann–Whitney test, $P = 0.0307$, Supplementary Fig. 8). Together, these results suggest that demyelination-induced changes in cytoplasmic Ca[2+] influx scale with the mt-Ca[2+] responses, and that Ca[2+] hot spots at branch points and low mt-Ca[2+] responses within the internodes are lost (Fig. 9).

## Discussion

We used a Cre-recombinase-mediated viral approach combined with single-cell reconstructions and found that myelinated segments of PV[+] axons have a high mitochondrial content (Figs. 4, 5). The increased number of internodal mitochondria is in striking contrast with the general notion that myelinated axons contain fewer mitochondria[16–19]. In a previous study based on saturated ultrastructural reconstructions of deep layers of the mouse neocortex it was estimated that the mitochondrial content in myelinated axons was a ~30-fold lower compared to unmyelinated axons[18]. However, in that study there was no cell type-specific differentiation. Using a publicly available EM database[38] we found that the mitochondrial distribution in intermittently myelinated axons differed between (PV[+]) basket cell interneurons and L2/3 PNs. In contrast to the mitochondrial heterogeneity in interneuron axons, mitochondria are homogeneously distributed across myelinated and unmyelinated segments in excitatory axons. Furthermore, cell-type-specific mitochondrial changes were also present following demyelination. While increased numbers of mitochondria were found in excitatory axons (Supplementary Fig. S5), consistent with a wealth of data in experimental demyelination and multiple sclerosis[7,10–13], demyelinated PV[+] axonal branches however showed reduced mitochondrial density.

The cellular and molecular mechanisms mediating myelin-dependent clustering of mitochondria to PV[+] internodes remain unknown. Somatostatin-expressing interneurons are also occasionally myelinated[20,25,64]. Cre-dependent approaches such as those presented in the current study and publicly available large-scale datasets[38] will help to elucidate the cell-type specific factors underlying

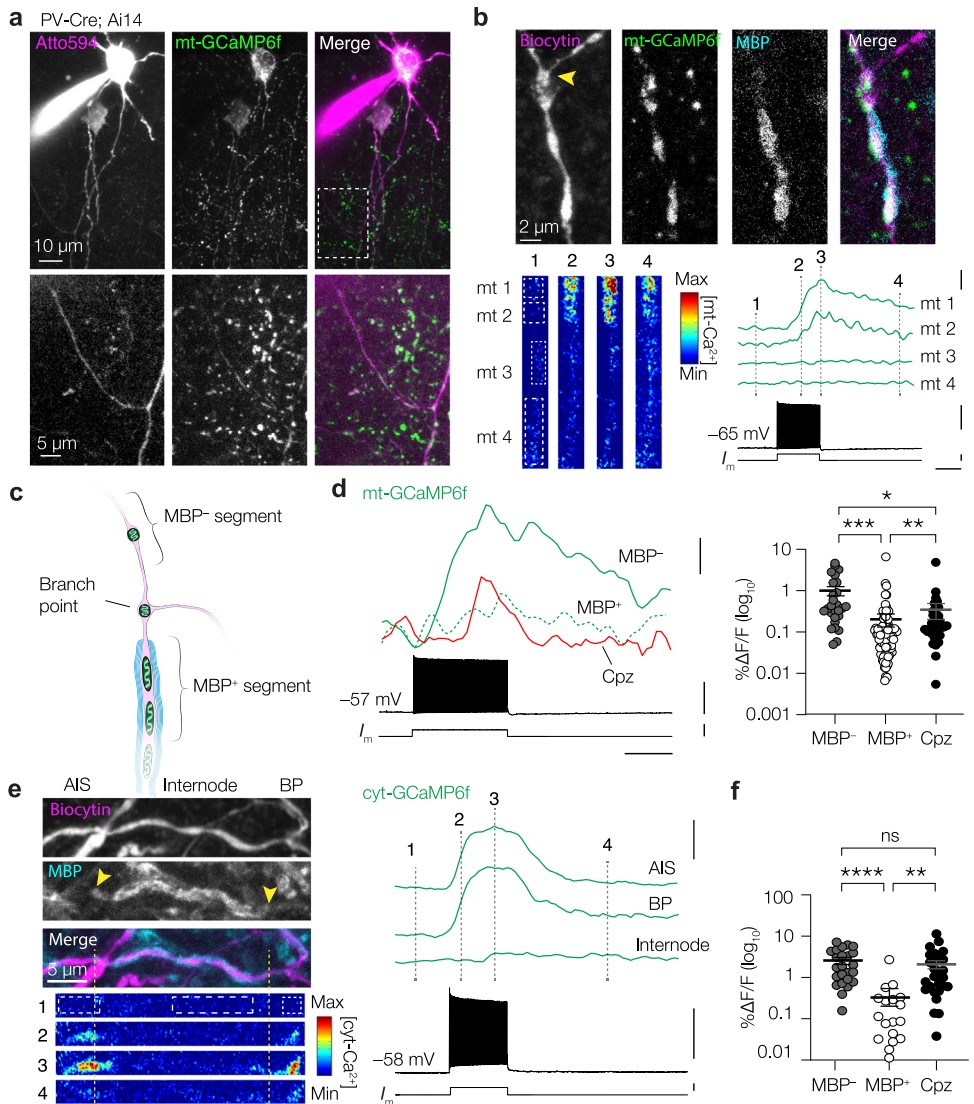

**Fig. 7 | Myelin attenuates internode mt-Ca²⁺ responses. a** Example 2P image of an Atto594-filled mt-GcaMP6f⁺ PV⁺ interneuron. Dashed box indicates the first branch point of the axon, with a characteristic large angle (≥90°). Similar results were obtained in 17 mice. **b** Example control PV⁺ axon with mt-GCaMP6f-labelled mitochondria. Mt-Ca²⁺ responses are stronger in MBP⁻ mitochondria (mito 1 and 2) compared to those in MBP⁺ segment (Mito 3 and 4). Dashed lines indicate time-points corresponding to the heatmap, yellow arrowhead indicates branch point. Scale bars indicate (top to bottom) 100% ΔF/F, 50 mV, 0.5 nA and 0.5 s. **c** Schematic overview of sampled axonal regions. In **d**, for cuprizone, axonal segments in between branch points were selected. Similar results were obtained in 17 mice. **d** Left, example Ca²⁺ responses of mitochondria in MBP⁺, MBP⁻ or putatively demyelinated segments. Right, stronger mitochondrial Ca²⁺ responses in MBP⁻ segments compared to MBP⁺ axon and increased Ca²⁺ responses after demyelination (Kruskal–Wallis test, $P < 0.0001$; Dunn's post hoc test MBP⁻ vs., MBP⁺ ***$P < 0.0001$; MBP⁻ vs. cuprizone, *$P = 0.0195$; MBP⁺ vs. cuprizone, **$P = 0.0039$;

MBP⁺, $n = 101$ mitochondria of 12 cells from 7 mice; MBP⁻, $n = 27$ mitochondria of 10 cells from 6 mice; cuprizone, $n = 36$ mitochondria of 10 cells from 5 mice). Scale bars indicate (top to bottom) 25% ΔF/F, 50 mV, 1 nA and 0.5 s. **e** Top: Example confocal image of the AIS, myelinated internode and first branch point of a PV⁺ axon Bottom: time lapse of a two-photon cytosolic (cyt-)GCaMP6f recording of the same axon in response to -100 APs. Notice only a slight Ca²⁺ response at the MBP edge but practically none in the rest of the internode. Scale bars indicate (top to bottom) 100% ΔF/F, 50 mV, 0.5 nA and 0.5 s. **f** Cyt-Ca²⁺ responses are stronger in unmyelinated and demyelinated axonal segments (Kruskal–Wallis test, $P < 0.0001$; Dunn's post hoc test MBP⁻ vs MBP⁺, $P < 0.0001$, MBP⁺ vs cuprizone, $P = 0.0004$, MBP⁻ vs cuprizone, $P = 0.4351$; MBP⁺, $n = 18$ segments of 6 cells from 3 mice; MBP⁻, $n = 24$ segments of 8 cells from 3 mice; cuprizone, $n = 28$ segments of 6 cells from 3 mice). Solid bars in **d** and **f** represent the mean, error bars indicate SEM. Individual data points in **d** represent mitochondria, individual data points in **f** represent segments. Source data are provided as a Source data file.

mitochondrial distribution. Interestingly, expression of the fast Ca²⁺ chelating protein parvalbumin alone most likely does not to explain clustering as in PV⁺ Purkinje axons mitochondria are homogenously distributed and demyelination increases mitochondrial density like in excitatory axons[12,13]. Locally projecting PV⁺ basket cell axons are small in diameter; on average -0.3 µm for unmyelinated and -0.6 µm for myelinated segments[25,65]. Small diameter axons possess a relatively large membrane surface/cytoplasm ratio limiting axial conductivity for AP propagation but may also require higher energetic costs for axonal

transport[63,66] and impede metabolic supply. The energetic demand of the small diameter PV⁺ interneuron axons is further augmented by the high-frequency firing rates and their generation of local gamma frequency oscillations[33]. Taken together, it is tempting to speculate that PV⁺ basket cell axon mitochondria efficiently consume the metabolic substrates supplied externally from oligodendrocytes. A direct external metabolic supply is consistent with the multiple distinctive features in their cellular and molecular organization compared to myelinated excitatory axons, including relatively high expression of mitochondrial

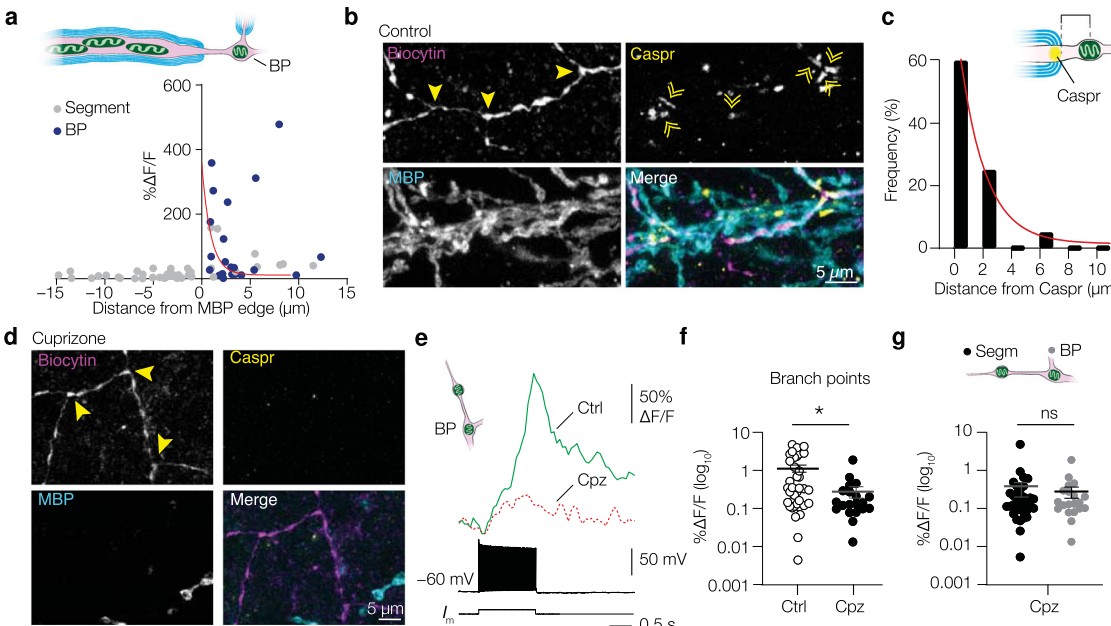

**Fig. 8 | The mt-Ca²⁺ responses are amplified at nodal domains in PV⁺ axons.**
**a** Relation between distances from myelin sheaths and mt-Ca²⁺ response ampli-tude. Red line indicates exponential decay fit ($y = (3.588 − 0.1117) \times \exp(−0.894 \times x) + 0.1117$); $n = 62$ mitochondria of 9 cells from 6 mice. **b** Example confocal images of PV⁺ axon branch points flanked by paranodes (Caspr). Similar results were obtained in 3 mice. **c** Mitochondria are frequently found close to paranodes. Red line indicates exponential decay fit ($y = (60.73 − 1.415) \times \exp(−0.5692 \times x) + 1.415$); $n = 20$ mitochondria of 4 cells from 3 mice. **d** Example of PV⁺ axon branch points of a cuprizone-treated mouse. Notice the overall lack of both MBP and Caspr immunostaining. Similar results were obtained in 3 mice. **e** Left: schematic repre-sentation of the compartments investigated in (**e**) and (**f**); Right: example traces

from control or cuprizone-treated axon branch point mitochondria. **f** Amplitude of branch point mt-Ca²⁺ responses was reduced upon demyelination (two-sided Mann–Whitney test, *$P = 0.0123$; control, $n = 38$ mitochondria of 11 cells from 7 mice; cuprizone, $n = 20$ mitochondria of 9 cells from 5 mice). **g** In demyelinated PV⁺ axons, mt-Ca²⁺ responses at branch points are indistinguishable from those in segments (two-sided Mann–Whitney test, $P = 0.8106$; segment, $n = 36$ mitochon-dria of 10 cells from 5 mice, same data as Fig. 7d; branch points, $n = 20$ mito-chondria of 9 cells from 5 mice, same data as in (**f**)). BP, branch point. Horizontal bars indicate means, error bars indicate SEM. Solid bars in **d** and **e** represent the mean, individual data points represent mitochondria. Source data are provided as a Source data file.

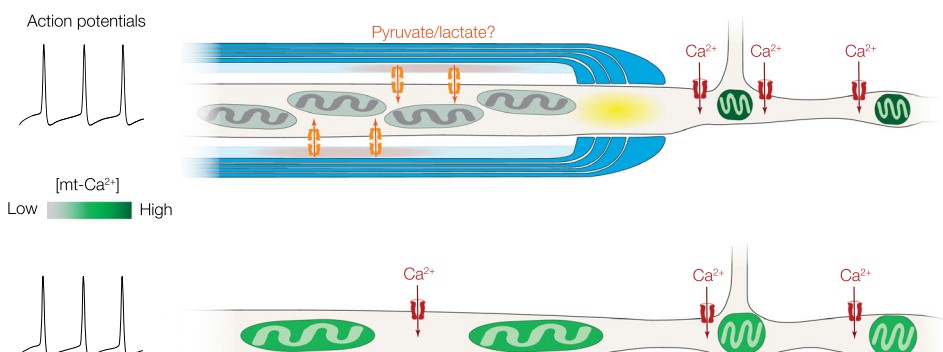

**Fig. 9 | Summary of the impact of myelin on PV⁺ axonal mitochondria.** In normally myelinated PV⁺ axons (top), mitochondria are clustered at high densities to myelinated internodes (blue), possibly due to local delivery of nutrients such as pyruvate or lactate (orange) from cytoplasmic noncompacted myelin (light blue) to the axon. Internodal mitochondria avoid the first -2 μm of the myelin sheath (paranodal domain, yellow), are relatively large, and display no or very weak

activity-dependent mt-Ca²⁺ transients (grey). In branch points and unmyelinated segments, mitochondria are smaller but AP-evoked mt-Ca²⁺ transients are large (dark green). Following myelin loss (bottom), mitochondria become larger and distribute uniformly along proximal axons. The mt-Ca²⁺ responses are uniform and low in amplitude (light green) and nodes of Ranvier are lost.

proteins in the axon (reviewed in ref. [67]). The myelin sheath of inter-neuron axons is characterized by higher levels of 2′,3′-cyclic nucleotide 2′-phosphohydrolase (CNPase)[37]. This enzyme is found in the cyto-plasmic channels in myelin and the noncompacted inner cytoplasmic loops, which may be involved in lactate and pyruvate supply to the periaxonal space, where these nutrients are shuttled into the axon via monocarboxylate transporters[3,4,6,37] (Fig. 9). Indeed, our findings in *Shiverer* mice indicate that noncompacted myelin wrapping sufficed to cluster mitochondria to internodes. Whether oligodendroglial lactate

and pyruvate supply acts as a trophic factor to immobilize mito-chondria remains to be tested. Interestingly, high glucose availability has been shown to negatively regulate mitochondrial motility[68]. Fur-thermore, live imaging in zebrafish axons showed that near paranodes there is myelin-dependent vesicle release[69,70], a process which requires high levels of ATP[71]. ATP consumption by vesicle release, the Na⁺/K⁺ ATPase[72] or axonal transport[73] might cause high local cytoplasmic adenosine diphosphate (ADP) levels, which are known to reduce mitochondrial motility[74] and thereby may potentially cluster

mitochondria to PV$^+$ internodes. An alternative mechanism for the mitochondrial clustering may be the absence of local activity-dependent cytoplasmic and mitochondrial Ca$^{2+}$ influx within the internode (Figs. 7, 8). Since mitochondrial ATP synthesis is in part regulated by Ca$^{2+}$ (see ref. 75), the mitochondrial clustering at internodes could be viewed as a compensatory mechanism to maintain a sufficient concentration of internodal ATP. To experimentally examine this requires directly imaging ATP within the PV$^+$ internodes and testing the contribution of internodal mitochondria to local ATP synthesis.

We cannot exclude the possibility that cuprizone affects mitochondria directly[76], but the increase in mitochondrial size upon demyelination (Fig. 3) is in keeping with previous studies in axons of other cell types and different models of demyelination[7,8,11], suggesting its toxicity does not play a role in the present analysis. To determine the features of motility, Ca$^{2+}$ buffering, anatomical properties of size and shape as well as the distribution of the mitochondria in PV$^+$ interneurons we used two-photon and confocal microscopy together with 3D EM data, which have been used previously to study mitochondria[7,9,35,50,77]. Confocal imaging has the advantage of allowing the mapping of mitochondria in large parts of single cells from multiple animals and distinct genetically modified mice but with the tradeoff of a lower spatial resolution. The EM data independently verified and confirmed our results at the ultrastructural level that mitochondria are larger and more densely distributed at the myelinated PV$^+$ axon. Moreover, the ultrastructural data enabled us to directly investigate mitochondria in different cell types from the same brain.

Our cyt-Ca$^{2+}$ data show that the heterogeneous mt-Ca$^{2+}$ responses along the PV$^+$ basket cell axons reflect local differentiation of the axon membrane. In normally myelinated axons we observed strong mt-Ca$^{2+}$ influx near branch points which were flanked by Caspr$^+$ signals, suggesting that branch points are sites of nodes of Ranvier, consistent with previous work in GABAergic axons[67,78]. At the nodal domains large amplitude sodium spikes are generated, opening voltage-gated Ca$^{2+}$ channels mediating local cytoplasmic Ca$^{2+}$ responses[57,58,79]. The lower Ca$^{2+}$ influx at these sites in both the axonal cytoplasm and mitochondria after demyelination might therefore reflect loss of nodal voltage-gated ion channel proteins and/or frequency-dependent failures during our trains of ~100 spikes. AP-evoked failures of release during PV$^+$-mediated inhibition of PNs has been reported during recording of synaptically coupled PV$^+$ basket cells and PNs[22]. However, during AP trains we still observed large mt-Ca$^{2+}$ transients in the downstream presynaptic terminals of the PV$^+$ axons (Supplementary Fig. 8). Our cyt-Ca$^{2+}$ imaging data suggest that mt-Ca$^{2+}$ transients at unmyelinated and putatively demyelinated segments are caused by AP-evoked membrane depolarizations opening voltage-gated Ca$^{2+}$ channels. Although we cannot exclude a role for presynaptic GABAergic terminals in proximal unmyelinated (MBP$^-$) opening voltage-gated Ca$^{2+}$ channels[50,59] such mt-Ca$^{2+}$ transients at release sites typically show much larger Ca$^{2+}$ responses (Supplementary Fig. 8). To better understand the biophysical properties of interneuronal myelination and the axolemmal membrane it will be critical to employ high temporal resolution voltage sensitive dye imaging and quantify absolute changes in membrane potential during axonal spiking[1,80].

The finding that myelin clusters mitochondria and defines mitochondrial Ca$^{2+}$ buffering (which also drives ATP synthesis[75]) indicates that myelin patterns may critically regulate energy homoeostasis in the PV$^+$ axon arborization. Such a link also has implications for interneuron myelin plasticity[81,82]. Recent work has revealed that experience-dependent myelin remodelling occurs prominently around PV$^+$ interneuron axons[82]. If in PV$^+$ interneurons newly formed myelin sheaths attract mitochondria this could represent a glial mechanism to strengthen axonal pathways and promote specific circuits of inhibitory transmission. Both myelination and mitochondrial distribution in PV$^+$ interneurons are known to be important for gamma oscillations, indicating they may represent converging pathways that shape PV$^+$

interneuron inhibitory functions[22,35]. More research will be necessary to test this hypothesis, by for instance using longitudinal mitochondrial imaging during myelin development or activity-induced myelination. Finally, the present findings may shed light on the vulnerability of PV$^+$ basket cells to myelin loss in MS[28–30]. If mitochondria in PV$^+$ axons depend on a glial source of metabolic support, myelin loss might acutely and negatively influence axonal transport, ion homoeostasis and other critical processes in PV$^+$ axons. Moreover, the small diameter of basket cell axons could make them more vulnerable to excess Ca$^{2+}$ influx upon demyelination[83], in particular given that mt-Ca$^{2+}$ buffering is insufficient to maintain normal cyt-Ca$^{2+}$ levels during AP trains (Fig. 7). Together, the data presented here reveal a critical role for interneuronal myelin to diversify mitochondrial distribution and function in PV$^+$ axons and encourage future research into mitochondrial dysfunction in PV$^+$ interneuron axons in neurodegenerative diseases.

## Methods
### Animals
All procedures were performed after evaluation by the Royal Netherlands Academy of Arts and Sciences (KNAW) Animal Ethics Committee (DEC) and Central Authority for Scientific Procedures on Animals (CCD, license AVD8010020172426). The specific experimental designs were evaluated and monitored by the Animal Welfare Body (IvD, protocols NIN19.21.01, NIN19.21.09 and NIN19.21.12). Male and female PV-Cre; Ai14, Rbp4-Cre, Rbp4-Cre; ChETA or PV-Cre × Ai14 × *Shiverer* mice were used in this study. PV-Cre × Ai14 mice were obtained by crossing B6;129S6-Gt(ROSA)26Sor$^{tm14(CAG-tdTomato)Hze}$/J (Jax strain 007908; RRID:IMSR_JAX:007908) with B6;129P2-*Pvalb*$^{tm1(cre)Arbr}$/J (Jax strain 008069; RRID:IMSR_JAX:008069). Rbp4-Cre × ChETA mice were obtained by crossing B6.FVB(Cg)-Tg(Rbp4-cre)KL100Gsat/Mmucd (MMMRRC strain 037128-UCD; RRID:MMRRC_037128-UCD) with B6;129-*Gt(ROSA) 26Sor*$^{tm1(CAG-COP4*E123T*H134R,-tdTomato)Gfng}$/J (Jax strain 017455; RRID:IMSR_JAX:017455). PV-Cre × Ai14 × *Shiverer* mice were obtained by crossing PV-Cre × Ai14 mice (see above) with C3Fe.SWV-Mbp$^{shi}$/J (Jax strain 001428; RRID:IMSR_JAX:001428). Mice were kept at a 12 h day-night cycle with access to food pellets and water ad libitum. Cages were open or IVC cages with corncob bedding. Wherever possible, animals were housed together with at least one cage mate. Ambient temperature was maintained at 20–24 °C, humidity at 45–65%. For cuprizone experiments, at 8 weeks of age, mice were fed powder food either with or without 0.2% cuprizone (biscyclohexane oxaldihydrazone, Sigma-Aldrich) supplement (range: 5–6 weeks). Animals on cuprizone diet that showed persistent >30% weight loss compared to control litter mates were excluded from the study. *Shiverer* mice were sacrificed before the phenotype prevented them from reaching food or water (persistent epileptic seizures, hind limb paralysis). The weight, locomotion and overall welfare of all mice were closely monitored.

### Plasmids
Cre-dependent mt-GFP and mt-GCaMP6f AAV plasmids were created by replacing the mCherry open reading frame (ORF) in pAAV-EF1a-mCherry-DIO (Addgene plasmid #20299, RRID: Addgene_20299) with mt-GFP or 4mt-GCaMP6f ('mt-GcAMP6f' in the main text). High-Fidelity Phusion Taq polymerase (Thermo Fisher Scientific, F-530XL) was used to amplify mt-GFP from Addgene plasmid #44385 (RRID: Addgene_44385) or 4mt-GCaMP6f (plasmid was generously donated by Diego de Stefani). For mt-GFP, PCR fragment and plasmid were digested with AscI and NheI (New England Biolabs). Because 4mt-GCaMP6f contains an NheI digestion site, to create the mt-GCaMP6f-DIO plasmid, pAAV-EF1a-mCherry-DIO was digested with NheI, blunted using T4 DNA Polymerase (New England Biolabs), digested with AscI and then dephosphorylated using Antarctic Phosphatase (New England Biolabs). The 4mt-GCaMP6f PCR product was phosphorylated

using T4 Polynucleotide Kinase (New England Biolabs) and digested using AscI. Plasmid and insert were then ligated overnight at 16 °C (T4 DNA ligase, New England Biolabs). After transformation into chemically competent *E. coli* and subsequent plasmid purification, insertion and double inverted orientation (DIO) were confirmed using restriction enzyme analysis and DNA sequencing. The plasmids were deposited to Addgene (mt-GFP-DIO #174112; 4mtGCaMP-DIO: #179529). To test Cre-dependent expression, HEK293-T cells (HEK 293T/17 cell line, CRL-11268 obtained from ATCC; RRID:CVCL_1926) were transfected with pAAV-EF1a-mt-GFP-DIO or pAAV-EF1a-mt-GCaMP6f-DIO only or pAAV-EF1a-mt-GFP-DIO and pCAG-Cre (Addgene #13775; RRID Addgene_13775) using polyethylenimine (PEI). Briefly, plasmid DNA (500 ng) was diluted in saline and then mixed with PEI, allowed to incubate for 20–25 min and then applied dropwise to HEK293-T cells in a 6-well plate. The next day using a fluorescence microscope, mt-GFP or mt-GCaMP6f expression was assessed, which was absent in the cells lacking or expressing only pCAG-Cre, confirming Cre-dependence.

mt-GFP primers: FW: 5′-AATAGCTAGCATCATGTCCGTCCTGACG-3′
RV: 5′-TAATGGCGCGCC GGAAGTCTGGACATTTATTTG-3′
4mt-GCaMP6f primers:
FW: 5′-ACT TCG CTG TCA TCA TTT G-3′
RV: 5′-GCTAGCCCACCATGTCCGTC-3′
Sequencing primers: FW: 5′-GGATTGTAGCTGCTATTAGC-3′
RV: 5′-GGCAAATAACTTCGTATAGGA-3′

## AAV production

HEK293-T cells of low passage (<25) were kept in Dulbecco's modified Eagle's medium (DMEM, Thermo Fisher Scientific #31966-047) containing 10% foetal calf serum (FCS, Thermo Fisher Scientific A4766801) and 5% penicillin-streptomycin (Pen-Strep, Thermo Fisher Scientific #15140122) at 37 °C and 5% $CO_2$. For virus production, cells were plated on 15 cm dishes at a density of $1–1.25 \times 10^7$ cells per dish (12 dishes per virus). The next day, medium was replaced with Iscove's modified Dulbecco's medium (IMDM, Sigma Aldrich I-3390) containing 10% FCS, 5% Pen-Strep and 5% glutamine (Thermo Fisher Scientific #25030081) 1–2 h before transfection. For transfection, AAV rep/cap, AAV helper (mt-GFP: AAV5; mt-GCaMP6f: AAV1) and transfer plasmids were mixed and diluted in saline before mixing with saline-diluted polyethylenimine (PEI, Polysciences #23966-2) and brief vortexing. After 20–25 min of incubation, the transfection mix was added in a drop-wise fashion to the culture plates. The next day, all medium was refreshed after which the cells were left for an additional two nights (i.e. 72 ho after transfection). Medium was discarded and cells were then collected and subsequently lysed with 3 freeze-thaw cycles to release AAVs. Cell lysate was then loaded on an iodixanol gradient (60, 40, 25 and 15% iodixanol, ELITechGroup #1114542) in Beckman Quick-Seal Polyallomer tubes (Beckman-Coulter #342414) which were sealed and centrifuged in a Beckman-Coulter Optima XE-90 Ultracentrifuge using a Type 70 Ti rotor at 16 °C and 69,000 rpm ($488727.6 \times g$) for 1 h and 10 min. The virus-containing fraction was extracted from the tubes and AAVs were then concentrated in Dulbecco's Phosphate Buffered Saline (D-PBS) + 5% sucrose using Amicon Ultra-15 (100 K) filter units (Merck Millipore UFC910024) at $3220 \times g$. At least 4 rounds of centrifugation were used to ensure complete replacement of iodixanol with D-PBS + 5% sucrose. Typical yields were ~150 μl of virus, the titre of which was determined using quantitative PCR (titres: mt-GFP-DIO, $4.43 \times 10^{13}$ gc/ml; mt-GCaMP6f-DIO, $3.44 \times 10^{13}$ gc/ml)[84]. Viral aliquots were stored at −80 °C until further use.

qPCR primers, recognize AAV2 inverted terminal repeats (ITRs):
5′-GGAACCCCTAGTGATGGAGTT-3′ 5′-CGGCCTCAGTGAGCGA-3′

## Viral injection

Viral injections were typically performed during the 3rd week of control powder food or cuprizone treatment. *Shiverer* mice were injected at 3–4 weeks of age. Mice were anaesthetized using isoflurane (3% induction, 1.2–1.5% maintenance) after which they subcutaneously received 5 mg/kg Metacam. Body temperature was monitored and maintained at 37 °C using a heating pad and eyes were prevented from drying out using eye ointment. The head was then shaved and placed in a stereotaxic frame (Kopf) and an incision was made in the skin along the midline. Lidocaine (10%) was administered on the periost before removing it. Small (<1 mm) bilateral craniotomies were made at −0.5 mm caudally from Bregma and 2.5 mm laterally from the midline without damaging the dura mater. With a sharp glass pipet attached to a Nanoject III (Drummond), 40–50 nL of virus was injected at 1 nL/s and at a depth of 450 μm. Approximately 3 min after finishing the injection, the needle was retracted slowly. Bone wax was applied to the craniotomies and the skin was sutured before mice were allowed to recover. Animals were monitored closely in the 3–5 days following surgery, during which their weight, locomotion and overall wellbeing were checked.

## Acute slice preparation

After 5–6 weeks of cuprizone or control treatment or, in the case of *Shiverer* mice, at 5–7 weeks of age (i.e. 2–3 weeks after virus injection), mice were deeply anaesthetized using pentobarbital (50 mg/kg, intraperitoneal injection) and transcranially perfused with ice-cold carbogenated (95% $O_2$, 5% $CO_2$) cutting artificial cerebrospinal fluid (cACSF; 125 mM NaCl, 3 mM KCl, 6 mM $MgCl_2$, 1 mM $CaCl_2$, 25 mM glucose, 1.25 mM $NaH_2PO_4$, 1 mM kynurenic acid and 25 mM $NaHCO_3$). After quickly dissecting out the brain 400 μm thick parasagittal slices were cut in ice-cold carbogenated cACSF using a Vibratome (1200S, Leica Microsystems). Slices were transferred to a holding chamber containing carbogenated cACSF where they were kept at 35 °C for 35 min to recover and then allowed to return to room temperature for at least 30 min before starting experiments.

## Electrophysiology and two-photon imaging

To perform electrophysiological experiments, slices were transferred to a recording chamber with continuous in- and outflow of carbogenated recording ACSF (rACSF, 125 mM NaCl, 3 mM KCl, 1 mM $MgCl_2$, 2 mM $CaCl_2$, 10 mM glucose, 5 mM L-Lactate, 1.25 mM $NaH_2PO_4$, and 25 mM $NaHCO_3$) at a rate of 1–2 ml per minute at 32 °C. Glass pipettes with an open tip resistance of 6–7 MΩ were filled with an intracellular solution containing (in mM, 130 K-Gluconate, 10 KCl, 10 HEPES, 4 Mg-ATP, 0.3 $Na_2$-GTP, 10 $Na_2$-phosphocreatine; pH -7.25, osmolality -280 mOsmol/kg) supplemented with 5 mg/ml biocytin (Sigma-Aldrich, B4261) and 50 μM Atto-594 (Sigma-Aldrich, A08637). Whole-cell recordings were made using a patch-clamp amplifier (Multiclamp 700B, Axon Instruments, Molecular Devices, RRID: SCR_018455) operated by AxoGraph X software (version 1.5.4; RRID: SCR_014284). Action potential trains were evoked using 700 ms step pulses, from −250 pA to +700 nA with increments of 50 pA. Single action potentials were evoked using 3 ms incremental (2.5–5 pA) step pulses with a starting amplitude below spike threshold. Using an AD/DA converter (ITC-18, HEKA Elektronik GmbH), voltage was digitally sampled at 100 Hz. The access resistance during current-clamp experiments (range: 15–30 MΩ) was fully compensated using bridge balance and capacitance neutralization of the amplifier. Somatic single-cell recordings were made from mt-GFP⁺ cells, which were visualized using a two-photon (2P) laser-scanning microscope (Femto3D-RC, Femtonics Inc., Budapest, Hungary). Imaging was controlled using MES software (Femtonics Inc., Budapest, Hungary, version 6.3.7902). A Ti:Sapphire pulsed laser (Chameleon Ultra II, Coherent Inc., Santa Clara, CA, USA) was tuned to 770 nm for two-photon excitation to visualize mt-GFP/mt-GCaMP6f and tdTomato. Fluorescent signals were detected using two photomultipliers (PMTs, Hamamatsu Photonics Co., Hamamatsu, Japan), one for mt-GFP and for Atto-594 (20 μM) to assess its diffusion, as an indication of biocytin diffusion. For motility

imaging, z-stacks were acquired every 10 s for 8–12 min. The Image Stabilizer plugin for FIJI was used to account for drift. For Ca²⁺ imaging, mt-GCaMP6f⁺ cells were targeted for single-cell patch clamping and filled with Atto-594 (50 μM; Sigma 08637) and biocytin (5 mg/ml; Sigma B4261). Next, z-stacks were made with the laser tuned to 800 nm to identify the axon using the Atto-594 signal. Axons were clearly distinguishable from dendrites, as they are much thinner, branch much more often and tend to project back towards the soma, as described previously[20]. Mitochondrial Ca²⁺ responses were then visualized with the laser tuned to 940 nm and recorded at ~20 Hz imaging frequency. Ca²⁺ responses were analysed using a custom-written Matlab script. Briefly, regions of interest containing mitochondria were selected from which background was subtracted, bleach correction and smoothing were applied, and ΔF/F was calculated. Care was taken to keep the number of APs, which were evoked using a 700 ms block pulse, the same between groups (MBP⁻, 105.6 ± 0.89 action potentials; MBP⁺, 103.6 ± 0.51 APs; cuprizone, 104.7 ± 1.06 APs, Kruskal–Wallis test $P = 0.1591$). MBP⁻ segments do not include AISs.

## Immunohistochemistry

For staining of biocytin-filled cells, upon completion of experiments acute slices were immediately placed into 4% PFA in PBS for 20–25 min, followed by 3 washes of 10 min with PBS. Next, tissue was blocked for 2 h in PBS containing 1–2% Triton-X and 10% normal goat serum. After blocking, sections were moved to blocking buffer containing primary antibodies (see Supplementary Table 1) and were incubated overnight at room temperature. Following three 10 min washes in PBS, sections were transferred to PBS containing secondary antibodies and were incubated overnight 2 h room temperature or overnight at 4 °C. The tissue was finally washed again in PBS three times for 10 min before mounting using FluorSave mounting medium (Merck-Millipore #345789). For immunostainings against MBP, the staining protocol was adjusted: blocking was done 1 h at 37 °C and 1 h room temperature and using 1% Triton, tissue was washed only once in PBS per washing step, and secondary antibodies were incubated 1 h at 37 °C and 1 h room temperature. For immunostainings on sections without biocytin-filled cells, 400 μm PFA-fixed sections were first cryoprotected by placing them into 30% sucrose-PBS solution until fully saturated. A sliding freezing microtome (Zeiss Hyrax S30; temperature controlled by a Slee Medical GmbH MTR fast cooling unit) was used to cut 40 μm sections, which were either placed in PBS for immediate use or stored at −20 °C in cryoprotectant solution (30% ethylene glycol, 20% glycerol, 0.05 M phosphate buffer) until further use. Immunostaining protocol was the same as for 400 μm sections, but the blocking buffer contained 0.5% Triton-X and incubation duration for secondary antibodies was 2 h at room temperature. All steps are performed with gentle shaking, with the exception of the 37 °C incubation steps. For analysis of normal and demyelinated subcortical white matter, Rbp4-Cre mice injected with AAV1-EF1a-mCherry-DIO and AAV5-EF1a-mtGFP-DIO (mixed 1:1) were first perfused with 1x PBS followed by 4% PFA-PBS. Brains were then dissected out and allowed to fix O/N in 4% PFA-PBS after the brains were cryoprotected using 30% sucrose-PBS and processed into 40 μm coronal sections as described above.

## Confocal microscopy

Imaging was performed using a Leica SP5 or SP8 X confocal laser-scanning microscope controlled by Leica Application Suite AF (version 3.5.7.23225). Biocytin-filled cells were imaged with a 63× oil-immersion lens using the tile-scan function with automated sequential acquisition of multiple channels enabled. Step sizes in the Z-axis were 0.3–0.75 μm, and images were collected at a 2048 × 2048 pixel resolution at 100–150 Hz. Mitochondria at AISs were imaged similarly, but only cells were imaged whose AIS was directed parallel to the imaging plane, and z-stack step sizes were 0.2 μm. Cellular reconstructions and

quantification of mitochondrial density and morphology in single cells or at the AIS were performed manually using Neurolucida Software (MBF Bioscience, version 2019.2.1 or 2020.1.3, 64 bit, RRID: SCR_001775). In PV⁺ interneurons, axons could easily be distinguished from dendrites based on their small diameter and extensive branching. Mitochondrial contours were drawn at the plane where the mt-GFP signal was brightest and was omitted when neurite branches deviated strongly from a parallel orientation with respect to the imaging plane (their location was still marked for density calculations). Mitochondria were counted as belonging to the reconstructed cell if the mt-GFP signal was located clearly inside the cytosol and followed the path of the reconstructed neurite. After completion of tracing, reconstructions and contours were analysed using Neurolucida Explorer (MBF Bioscience, 2019.2.1, RRID: SCR_017348), which calculated contour area, aspect ratio (length:width), density, axonal length and inter-mitochondrial distance. For analysis of mitochondrial densities at the myelin sheath, the MBP signal was turned off during tracing of PV⁺ axons and marking their mitochondria, and vice versa. FIJI (RRID: SCR_002285) was used to extract partial images for use in figures. After preprocessing in FIJI, (FIJI 64 bit; ImageJ version 1.53q; Oxford Instruments, version 9.7.2, RRID: SCR_007370) was used to generate three-dimensional surface renderings. The spot functionality of Imaris was used to count mitochondria in the field of view for motility analysis. Motile mitochondria were identified by eye, mitochondria were considered stable if they displaced less than 2 μm.

## 3D Electron microscope data analysis

3D EM data was obtained from the Microns dataset[38] (www.microns-explorer.org), which entails a 1 mm³ EM block of primary visual cortex of one P87 male mouse. Basket interneurons were identified based on their morphology, i.e. their relatively round soma (as opposed to a pyramidal shape), the lack of spines on dendrites and a thin and highly branched axon that was partly myelinated. From each cell, one to two volumes were selected that either contained an axonal branch point with one myelinated and one unmyelinated branch, or a single branch with intermittent myelination. The EM volume of interest and cytosolic segmentation were then downloaded, after which the mitochondria were segmented and the segment length traced manually using Volume Annotation and Segmentation Tool (VAST) software (version 1.4.1)[85]. Because we found that branch points in PV⁺ interneurons are often nodes of Ranvier (a specialized axonal compartment), branch points and mitochondria that resided in them were not included in the analysis. Boutons were excluded for the same reason. If a mitochondrion spanned a border between myelinated and unmyelinated axon or beyond a branch point, they were not included. To analyse the mitochondrial content in L2/3 PNs, we used the same approached as described above. These cells were readily identified based on their pyramidal shape, spiny dendrites and distance from the pia. We selected L2/3 PNs that displayed patchy myelin. Fully or unmyelinated L2/3 PN axons were not included. Length and volume measurements were performed and exported using VAST Tools Matlab scripts. For examples used in figures, 3D models were exported as OBJ files using VAST software and rendered using 3ds Max (Autodesk, version 25.0.0.997, SCR_014251). See Supplementary Table 2 for cells used in the analysis.

## Statistics and reproducibility

All statistical comparisons were done using Prism (Graphpad, version 8.4.3, RRID: SCR_002798). Normality of datasets was determined using D'Agostino & Pearson or Shapiro-Wilk tests. We applied non-parametric tests if data deviated significantly from a normal distribution. For comparisons between groups we used a two-tailed unpaired t-test (normal data) or a two-tailed Mann–Whitney test (non-normal data). For t-tests degrees of freedom (df) are reported in the figure legends. One-way ANOVA (normal data) or Kruskal–Wallis test

(non-normal data) was applied for comparisons of three groups; two-way ANOVA was used to test interactions between groups and treatments; Bonferroni's post hoc test (normal data) or Dunn's post hoc test (non-normal data) was used for multiple comparisons; nested t-tests or nested one-way ANOVAs were used when large numbers of datasets were involved (i.e. mitochondrial contours) to avoid overpowering of non-nested statistical tests (cells were nested inside their respective treatment groups). Figure legends contain *P*-values and *n*-numbers and whether the latter signify mice, cells or mitochondria. Means are presented with SEM.

Data was collected from multiple cells from multiple animals to ensure reproducibility. Experiments were replicated in a minimum of 4 cells from a minimum of 3 mice with the exception of the 3D EM reconstructions which were done using data obtained from one mouse. For exact *n* numbers see figure legends. Reconstructions and recordings from different subcellular compartments of the same cell were obtained wherever possible. Age-matched mice were randomly allocated to either the control or cuprizone treatment. For comparisons of subcellular compartments (e.g. myelinated vs unmyelinated axon, axon vs dendrite, segment vs branch point), randomization is not possible as the group data points belong to are determined by their subcellular localization. Therefore, randomization was not performed in these cases. However, selection of cells for patching prior to e.g. cellular reconstruction and calcium imaging was done randomly (i.e. virally transduced cells with a healthy appearance were targeted but otherwise no selection was made). Blinding was not performed in this study as the difference between control and cuprizone-treated brain tissue (or brain tissue from Shiverer mice) can be readily observed during the performance of the experiments (i.e. due to the absence of myelin). In control cells, calcium imaging was done blind to the myelination state of axonal segments, which was revealed only later by immunohistochemistry. To determine the density of mitochondria in MBP⁺/MOG⁺ and MBP⁻/MOG⁻ segments, during tracing of the axon and mitochondria the MBP or MOG channel was disabled and vice versa. This was not possible in the 3D EM dataset, where the presence of myelin is readily observed in the image data that is required to perform the segmentation. Electrophysiological data was excluded if cells had an unstable resting membrane potential between sweeps. Calcium data was excluded when the number of action potentials deviated by more than 15% (<85 or >115 APs) or if the subsequent immunostaining was not reliable.

### Reporting summary
Further information on research design is available in the Nature Portfolio Reporting Summary linked to this article.

## Data availability
All data are available upon request to the corresponding authors. The Microns dataset used for 3D EM analysis is available at www.microns-explorer.org. Source data are provided with this paper.

## Code availability
Custom-written Matlab scripts for $Ca^{2+}$ imaging analyses along with pseudocode and example data can be found at https://github.com/Kolelab.

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

## Acknowledgements

The authors are thankful to Fred de Winter and Joost Verhaagen (NIN–KNAW) for the support for developing and producing the AAV constructs. We thank Christian Lohmann for critical reading of the manuscript and feedback during this research. We thank Diego de Stefani for generously sharing the 4mt-GCaMP6f plasmid with us. We thank Christiaan Levelt for generously sharing the AAV1-CAG-Flex-mRuby2-GSG-P2A-GCaMP6f virus with us. We thank Arnoldo Zaldivar Castro for optimizing the immunohistochemistry protocol. This study was, in part, supported by The Dutch Research Council (NWO, Vici 865.17.003) to M.K., a ZonMW Off Road grant 04510012010066 to K.K. and an Erasmus scholarship G ATHINE 01 to M.E.B.

## Author contributions

Conceptualization: M.K. and K.K. Methodology: M.K., K.K., M.E.B. and B.V. Investigation: K.K., M.E.B., B.V. and N.P. Formal analysis, M.K., K.K., M.E.B. and B.V. Visualization: M.K. and K.K. Funding acquisition: M.K. and K.K. Project administration: M.K. Supervision: M.K. and K.K. Writing—original draft: K.K. Writing—review & editing: M.K. and K.K.

## Competing interests

The authors declare no competing interests.
