## [Peer Review File · Nature Communications]

Parvalbumin basket cell myelination accumulates axonal mitochondria to internodesReviewers' Comments:

Reviewer #1:

Remarks to the Author:

In this research paper, Kole and colleagues report novel Cre-dependent AAV vectors allowing mitochondria and mitochondrial calcium changes to be investigated. Using these tools, they find that mitochondria cluster to myelinated segments of PV+ axons and that mitochondria at the AIS and in axons of PV+ interneurons are larger in size following demyelination. Mitochondria are lost in proximal branches upon demyelination suggesting that myelin wrapping clusters mitochondria locally within PV+ internodes. Two-photon imaging of action potential-evoked mitochondrial calcium (mt-Ca²⁺) responses in brain slices revealed high mt-Ca²⁺ in unmyelinated segments and branch points, while demyelination caused uniform low mt-Ca²⁺ responses. The authors suggest that myelin dampens mt-Ca²⁺ responses and that demyelination results in a more uniform mt-Ca²⁺ buffering profile in control PV+ axons.

Overall, the paper shows experiments of reasonable quality and the new AAV tools appear to have a good potential to advance knowledge on mitochondrial function. While these vectors could be used for both ex vivo and in vivo imaging studies, it would have been great to see some in vivo two-photon imaging as well that could strengthen the points of the authors concerning myelination states and mitochondrial calcium dynamics of the axons they have investigated. I also feel that with some more work that authors could substantially increase the impact of these studies and make their conclusions stronger.

Major points:

- It remains unclear from the present study to what extent myelination itself is responsible for mitochondrial clustering, volume changes and what the underlying molecular mechanisms may be. Cuprizone is a copper chelator that has been found to reduce complex IV of the mitochondrial respiratory chain in Purkinje neurons and toxicity on isolated mitochondria, neurons and glial cells has also been noted in previous studies (e.g. Varhaug et al., *Mitochondrion* 2020; Faizi et al., *Toxicol Mech Methods* 2016; Martínez-Pinilla et al., *Brain Sci* 2021; Luo et al., *Front Neuroanatomy* 2020). Thus, links between impaired myelination and mitochondrial function remain largely correlative even if the authors quantitatively assess that myelination-induced changes on mitochondria are not uniform along PV+ axons.
- Either using acute slices or genetic models, the authors could apply more direct approaches to target oligodendrocytes and study how mitochondrial functional properties are altered. Complementary models of demyelination also exist. As such, the statements that (de)myelination impacts on mitochondria rely solely on the effects of cuprizone and in the absence of specific molecular mechanisms the functional implications of these observations remain unclear.
- The authors show that activity-dependent mt-Ca²⁺ buffering in presynaptic terminals and the AIS of PV+ interneurons is unaffected by myelin loss. However, mt-Ca²⁺ transients in cuprizone-treated axons were stronger compared to MBP+ control segments and demyelination reduced mt-Ca²⁺ responses in PV+ axonal branch points. I wonder why the authors have not tried to influence oligodendrocyte-axon metabolic coupling, gap junctions etc by using pharmacological tools to study how this alters mt-Ca²⁺ responses in PV+ axonal branch points and in the AIS, for example. Such experiments could provide a way more direct evidence to links between the myelination and activity of mitochondria.
- Fig.1c: Please provide a high resolution insert showing colocalization of mt-GFP with TdTomato.
- Fig.3: The material presented in this figure is mostly methodological, it would be better suited as a supplementary figure considering other similar observations published previously.
- Fig.2e and 4i: Lack of significant effect of cuprizone on the number of mitochondria may be due to low n numbers.
- Fig.5a and d: Representative confocal images validating the drawings must be provided.

- Fig.6 and Fig.7: I miss a deeper characterisation of the mt-Ca²⁺ sensor here and the specificity of mitochondrial- and myelin-related interventions on the mt-Ca²⁺ signal. Already, it would be easier to appreciate Fig.6b, if time lapse images of given mitochondria would be provided and the dynamic range of mt-Ca²⁺ signal over time would be depicted. In functional terms, how does AP-induced mt-Ca²⁺ signal propagates in nearby mitochondria, would this be impaired in genetic models of myelin loss, by connexin blockers or in Cx47 or Cx32 KO mice? Would interfering with mitochondrial function directly show up in the mt-Ca²⁺ signal and how would this be affected in control mice or after demyelination?

Reviewer #2:

Remarks to the Author:

Axons of parvalbumin +ve inhibitory interneurons in the cerebral cortex are extensive and have a complex arborisation. They are partially myelinated and the patterning of myelin is influenced by the shape of the axonal arborisation, such that sheaths tend to be more frequent proximally and are absent at branch points and en passant boutons. The function of myelin on Pv +ve interneurons is not fully understood. Here, the authors aim to explore the proposal that myelination of Pv +ve interneuron effects trophic support of axonal mitochondria to maintain the cells' very high levels of spike generation and activity during gamma oscillation. They demonstrate in murine somatosensory cortex that myelinated segments of Pv +ve cortical interneurons have a higher mitochondrial clustering (mitochondria per unit length) than equivalent non-myelinated or (putative) demyelinated segments; that mitochondria at branch points and nodes of Ranvier more strongly buffer AP-evoked calcium influx than mitochondria elsewhere; and that experimentally induced myelin loss, leads to uniformly low AP-induced calcium uptake by axonal mitochondria. Together, these data demonstrate a role for myelin on Pv +ve interneurons in shaping the distribution and function of mitochondria. They do not necessarily shed light on the proposal that the function of myelin on Pv +ve interneurons is to effect trophic/metabolic support of axonal mitochondria.

The work builds on two main lines of research. The first being in relation to the influence of myelin, and de- or dysmyelination on the distribution of mitochondria in axons as well as other aspects of axonal energy supply and demand; the second being in relation to the function of myelin on Pv +ve interneurons and the susceptibility of these cells to injury in MS. The work is original and provides new information in relation to both aspects.

The results are compatible with the conclusion that "myelin [or functional consequences of myelination] clusters mitochondria and diversifies mitochondrial Ca²⁺ buffering" on Pv +ve interneurons. It is interesting to speculate that the findings "indicate[s] that myelination at the level of branch orders may provide an axoglial mechanism to regulate energy homeostasis in PV+ axons", but this. One might imagine that if mitochondrial "fuelling" occurs beneath myelin, these would subsequently redistribute to non-myelinated regions, but the authors demonstrate that movement is very limited, at least within the time frame examined.

The analyses and methodologie appears sound, with the caveat that I have no personal experience of AAV or reporters of calcium buffering. Limitations are described in the discussion and n values and statistical analyses are clearly indicated. The methods are clearly described, and primary data are presented illustrating some of the critical aspects of the protocols, such as the spatial distribution of the reporters upon administration into the brain. As the results are in contrast to previous observations in white matter and/or glutamatergic neurons, where mitochondrial volume is increased in non-myelinated regions of otherwise myelinated axons (e.g. in the optic nerve) it would seem appropriate for the authors to show they can replicate previous work, using the method adopted here. They may already have data from L5 PNs or could incubate ex vivo optic nerve, including the non-myelinated nerve head, with AAV.

If feasible, it would be interesting to know whether increased mitochondrial clustering is dependent upon compact myelin being present, or whether ensheathment per se is sufficient. This should be possible in shiverer mice and would help determine whether insulation (a function of compact

myelin) is required for mitochondrial clustering.

Minor points:

Perhaps the authors could speculate a little more on how their observations may shed light on the susceptibility of PV+ interneurons in multiple sclerosis.

Intro Lines 62-64 Consider citing PMID: 16555298, which describes the distribution of mitochondria in axons that are never surrounded by compact myelin, in shiverer mice, which have otherwise healthy axons, and 14736793, which examines mitochondrial distribution in myelinated and non-myelinated regions of the healthy adult human optic nerve.

"being partially electrically separated from the myelin sheath" meaning what?

Result Line 07. It might be clearer to mention the specific Cre driver lines at this point.

Results Lines 66-67. What is the evidence that 5 mM L-lactate and 10 mM glucose resembles the in vivo cellular environment?

Page 14, lines 34-35. Suggest changing "Myelin loss is typically associated with" to "Myelin loss can lead to"

Is the graph in 4K a cumulative distribution plot?

Page 17, "These data suggest that in PV+ axons, branch points are sites of nodes of Ranvier and mitochondria are clustered to the nodal domains." Consider citing PMID 18381760, which shows that in small diameter myelinated axons in white matter, there appears to be no clustering of mitochondria at paranodal/nodal regions of small diameter fibres.

Figure 7C, does this refer to a specific population of mitochondria only?

Discussion, line 71 "...but also impedes transport and metabolic supply" seems not entirely clear.

Reviewer #3:

Remarks to the Author:

In this paper, Kole et al. study, how in parvalbumin-positive (PV) interneurons myelination affects mitochondrial morphology, distribution and function – under normal conditions and during cuprizone-induced demyelination.

While a number of studies have addresses the relationship between myelination and mitochondria, e.g. in the optic nerve, this study targets an inhibitory neuron population with substantially, but still partially, myelinated axons in gray matter. Given the importance of PV neurons in cortical function, their unique firing patterns and the experimental asset of being able to compare myelinated and non-myelinated segments of the same axon, this is an interesting model. Indeed, overall I appreciate the chosen experimental model, the combination of techniques that integrates electrophysiology in the characterization of the neurons (which is missing in many other comparable papers) and the care with which the data are reported, quantified and illustrated. Also, the results suggest that PV interneurons might indeed differ in some aspects of their mitochondrial distribution and demyelination response from the excitatory neurons analyzed so far. Still, the paper left me unsatisfied with the level of new mechanistic insight actually gained. A substantial part of the paper shows controls or confirms known facts (Fig 1-3 do not add too much to the core of the story and could actually be supplementary: Fig 1 shows technical matters that are not too surprising, Fig 2 data that do not really relate to mitochondria in the end and Fig 3 shows facts established for many neurons already). At the same time, the more innovative parts (local density myelination vs. non-myelinated segments, demyelination reaction, the mitochondrial calcium dynamics) in my view only paint a partial picture open to many interpretations (falling short of the level of analysis achieved in previous work, where e.g. dynamic analyses of mitochondria were done or factors that determine local mitochondrial anchoring were probed, which should operate differently here). So I feel that substantial data and analyses would be needed to suitably revise this paper for an audience that is not very specialized; the authors address many of these points in

the discussion, but several are within their technical reach, so it is not clear why they chose to speculate rather than measure.

Major points:

1) A weakness is in my view the limited (experimental) analysis of the calcium imaging data. The authors write about 'calcium buffering', which has an interpretative quality, suggesting that mitochondrial calcium uptake here is serving to stabilize ("buffer") calcium levels elsewhere – in the discussion it then seems that the authors are more concerned with the metabolic regulation that such different calcium levels or peaks could imply (which is rather linked to the calcium-dependence of some key metabolic enzymes, not a possible "buffering" function). However, in the end, the authors only measure the levels of free calcium in the mitochondrial matrix, which essentially represents the sum outcome of a number of features – mitochondrial and not - that they do not quantify. As the authors mention in the discussion, mitochondrial calcium levels should be governed by essentially similar forces as any ion equilibrium, i.e. driving force (calcium levels in and around the mitochondria plus electrochemical gradient across the inner mitochondrial membrane) and conductance (mostly MCU). So, indeed, if cytoplasmic calcium level fluctuations differed between myelinated and non-demyelinated segments/ branch-points, then different levels of mitochondrial calcium influx would be expected even if the mitochondria were essentially identical in their makeup, e.g. with regards to membrane potential or calcium channel composition and density. But it remains unclear to me why the authors undertake no experimental effort to explore these different possibilities, e.g. by parallel measurements of cytoplasmic calcium levels around the respective mitochondria populations (which should be technically feasible), but perhaps also estimates of mitochondrial potential (which might be more difficult, but at least such measurements have been reported in some in vivo systems – and might be more precise in a systems where they can be internally normalized by comparing mitochondria in different parts of the same axon).

2) The authors make the intriguing observation that the proximal axon segments gain length during cuprizone-induced demyelination. This is surprising, as it seems hard to imagine how this could work simply given the density of cortical neuropil; the prediction would be that these axon segments are more meandering than their normally myelinated counterparts. But irrespective of this – it seems to me that this could explain a substantial part of the changes in density, i.e. that much of the actual change induced by demyelination could be axonal (such as intercalated growth e.g. by lateral vesicle fusion, or release of some form of longitudinal tension), not mitochondrial. Overall it seems that in Fig. 4 the total number of mitochondria per PV axon is the most stable value (although it is not directly reported). While of course for the measure of mitochondrial density, changes in nominator or denominator have essentially the same effect, cell biologically and mechanistically it makes a big difference, whether axons locally expand in length or mitochondria are locally added or removed. So these options should be experimentally distinguished.

Taking these two points together, it seems like many of the described changes could be explained purely by changes in axonal membranes and conductivity, with little need to evoke a true change in mitochondrial behavior – and it might be axonal, not mitochondrial features that is peculiar to PV neurons compared to the axons thus far analyzed in the literature, or to myelinated vs. non-myelinated segments, and mitochondria might just appear different, because these differences manifest in their calcium transients or density. I think to add a mechanistic dimension to their observations, distinguishing these options in a clear way would be important.

3) The observation of increased mitochondrial density under myelinated vs. non-myelinated segments is indeed notable and counterintuitive, given that both features are often interpreted as representing alternate adaptations to deal with high energy demand in fast-firing axons – and in this case both classes of axon segments would essentially have to sustain the same firing pattern, with the myelinated-segments being much better equipped to handle the resulting bioenergetic demands. However, it has been well appreciated that within a myelinated axon segment the mitochondrial distribution often is not uniform (e.g. with low density in the center of internodes and the nodes themselves, but relative accumulations in the para/juxta-nodal areas). So it would be informative to know how mitochondria are distributed within the myelinated and non-myelinated areas, and whether the density changes systematically at a border area (e.g. on the

two sides of a full node, the myelinated vs. non-myelinated side of a heminode etc.). After all, it remains somewhat unclear how local the effects of a mitochondrion as an ATP source are, and how far e.g. ATP, glucose, lactate or other small metabolites could diffuse inside the axoplasm. In the discussion, the authors seem to argue that perhaps the prevailing assumptions about energy demand in internodes are not universally true – and based on the difference between this report on PV axons and the axon populations previously analyzed (e.g. in the optic nerve) the suggestion is that different axon classes have different local energy demands. The authors e.g. point to axon diameter and surface/volume ratios and sub-myelin ion transients. Can this be analyzed, e.g. by looking for additional local correlations in axon diameter and mitochondrial density? Would it be possible to locally measure ATP levels and dynamics using a biosensor such as ATEAM (as done by the Nave lab in optic nerve axons)?

4) The authors openly discuss the shortcoming of their analysis using conventional light microscopy, which is laudable. At the same time, it seems to me that given the availability of annotated large scale 3D EM data sets, it should be possible to confirm some results in Fig. 5 experimentally at ultrastructural resolution.

Minor comments:

Fig 1c vs f: It would be more comparable to show in both cases (PV and Rbp4) the Ai-labeled population and/or a filled single neuron – ideally both in three channels.

Fig. 1d not really necessary

We are grateful for the reviewer points and have thoroughly revised the manuscript to reflect their comments. Here below the reviewer comments in black and authors responses in blue.

Reviewer #1 (Remarks to the Author):

In this research paper, Kole and colleagues report novel Cre-dependent AAV vectors allowing mitochondria and mitochondrial calcium changes to be investigated. Using these tools, they find that mitochondria cluster to myelinated segments of PV+ axons and that mitochondria at the AIS and in axons of PV+ interneurons are larger in size following demyelination. /... / Overall, the paper shows experiments of reasonable quality and the new AAV tools appear to have a good potential to advance knowledge on mitochondrial function. While these vectors could be used for both ex vivo and in vivo imaging studies, it would have been great to see some in vivo two-photon imaging as well that could strengthen the points of the authors concerning myelination states and mitochondrial calcium dynamics of the axons they have investigated. I also feel that with some more work that authors could substantially increase the impact of these studies and make their conclusions stronger.

We thank the reviewer for their positive comments on the potential of the viral tools. Although we have not yet performed in vivo imaging of mitochondria in the course of cuprizone-induced demyelination, which indeed could provide valuable insights (e.g. Nikic *et al.* (2011) *Nat Med* 17, pg. 495-499) we believe that with the additional experimental data we have strengthened the support for our main conclusion.

Major points:

- It remains unclear from the present study to what extent myelination itself is responsible for mitochondrial clustering, volume changes and what the underlying molecular mechanisms may be. Cuprizone is a chopper chelator that has been found to reduce complex IV of the mitochondrial respiratory chain in Purkinje neurons and toxicity on isolated mitochondria, neurons and glial cells has also been noted in previous studies (e.g. Varhaug *et al.*, *Mitochondrion* 2020; Faizi *et al.*, *Toxicol Mech Methods* 2016; Martínez-Pinilla *et al.*, *Brain Sci* 2021; Luo *et al.*, *Front Neuroanatomy* 2020).

There is indeed some evidence that cuprizone affects the mitochondrial enzyme and respiratory functions. We agree with the reviewer that we cannot exclude the possibility that cuprizone has a direct effect on neuronal mitochondria and added into the revised manuscript a comment on this caveat (“We cannot exclude the possibility that cuprizone affects PV+ IN mitochondria directly”, pg. 27). Nonetheless, the mitochondria enlargement is consistent with other studies of MS or other models of demyelination (e.g. Zamboni, J. L. *et al.* (2011) *Brain* 134, p. 1901–1913; Witte, M. E. *et al.* (2009) *J Pathology* 219, p. 193–204; Kiryu-Seo *et al.*, (2010) *J Neurosci* 30, p. 6658-66; Ohno *et al.* (2014) *Proc National Acad Sci* 111, p. 9953–9958).

Thus, links between impaired myelination and mitochondrial function remain largely correlative even if the authors quantitatively assess that myelination-induced changes on mitochondria are not uniform along PV+ axons.

We respectfully disagree with the reviewer. It is a strength of our approach that we quantify mitochondria along the axonal branches of PV interneurons *within* control interneurons as well as a *between* groups. To the best of our knowledge, it is the first demonstration that the presence of myelin predicts the local mitochondrial content. In the revised version additional independent support for this

conjecture has been added from 3D EM blocks (new Fig. 5, pg. 16). These data show that such a direct relationship between myelin and axonal mitochondria only exists for PV interneuron axons but not in pyramidal neuron axons. Secondly, we go beyond correlations by performing demyelination as an interrogation experiment. These results indicate that demyelinated branches have lower number of mitochondria (Fig. 3). Since no loss in mitochondria numbers was found outside of the proximal axonal branches, the argument that we are dealing with a general cellular toxicity by cuprizone seems not relevant. Furthermore, consistent with findings in white matter MS tissue or cuprizone-mediated demyelination (Zamboni, J. L. *et al.* (2011) *Brain* 134, 1901–1913; Ohno *et al.* (2014) *Proc National Acad Sci* 111, 9953–9958), Rbp4-Cre axons in the corpus callosum show in contrast to PV interneuron internodes an *increase* in mitochondria number (new Supplementary Fig. 5, Supplementary Information pg. 7). Finally, a third line of evidence comes from the triple transgenic mouse line (new Fig. 6, pg. 18). Together, we believe these experimental data argue for causality.

- Either using acute slices or genetic models, the authors could apply more direct approaches to target oligodendrocytes and study how mitochondrial functional properties are altered. Complementary models of demyelination also exist. As such, the statements that (de)myelination impacts on mitochondria rely solely on the effects of cuprizone and in the absence of specific molecular mechanisms the functional implications of these observations remain unclear.

As suggested by the reviewer we have generated a triple transgenic mouse line to study myelination in PV interneurons (PV-Cre x Ai14 x Shiverer). Shiverer mice have a mutation in *Mbp* lacking the compact myelin but preserving noncompact myelin wrapping (Roach *et al.*, *Cell* 42:149–155, 1985; Rosenbluth *J Comp Neurol* 194:639–648, 1980; Snidero *et al.* *Cell* 156:277–290, 2014). Interestingly, these new data (new Fig. 6, pg. 18) confirmed the heterogeneity in mitochondria content along myelinated and unmyelinated branches. Also in the shiverer mice we observed a high mitochondria content in myelinated axons indicating that noncompact myelin suffices. Together, these data provide independent (non-cuprizone-related) evidence for clustering.

- The authors show that activity-dependent mt-Ca²⁺ buffering in presynaptic terminals and the AIS of PV+ interneurons is unaffected by myelin loss. However, mt-Ca²⁺ transients in cuprizone-treated axons were stronger compared to MBP+ control segments and demyelination reduced mt-Ca²⁺ responses in PV+ axonal branch points. I wonder why the authors have not tried to influence oligodendrocyte-axon metabolic coupling, gap junctions etc by using pharmacological tools to study how this alters mt-Ca²⁺ responses in PV+ axonal branch points and in the AIS, for example. Such experiments could provide a way more direct evidence to links between the myelination and activity of mitochondria.

We are glad the reviewer noted that the activity-dependent mitochondria Ca²⁺ uptake was not affected in terminals and the AIS, which further argues against a general neuronal mitochondria toxicity by cuprizone. It is unclear to us which pharmacological approaches the reviewer is referring to. We show that the presence of the myelin limits or prevents activity-dependent mt-Ca²⁺, consistent with the low cytosolic Ca²⁺ responses in the myelinated PV internodes. Please note, new results in the revised manuscript (new Fig. 7, pg. 21) show that PV myelination also attenuates the cytoplasmic calcium influx.

- Fig.1c: Please provide a high resolution insert showing colocalization of mt-GFP with TdTomato.

We thank the reviewer for this suggestion. We have created new high-resolution images showing the co-localization of tdTomato and mt-GFP (new Fig. 1c, e, pg. 6). Quantifications of the overlap between mitoGFP and tdTomato are now mentioned in the main text (pg. 4)

- Fig.3: The material presented in this figure is mostly methodological, it would be better suited as a supplementary figure considering other similar observations published previously.

We agree with the Reviewer and moved this figure to the Supplementary materials (now Supplementary Fig. 3, Supplementary Information pg. 4)

- Fig.2e and 4i: Lack of significant effect of cuprizone on the number of mitochondria may be due to low n numbers.

We agree that this may be the case. We have compared data from 8 versus 9 axons which may be too low in statistical power. However, the myelination pattern of PV⁺ axons is highly heterogeneous which may be a confounding factor when comparing populations data for demyelination-induced changes. One refinement is the branch-order specific analysis (new Fig. 3i, pg. 11) and, secondly, we have now more directly studied the local mitochondria density using triple immunofluorescence staining (Fig. 4, pg. 13).

- Fig.5a and d: Representative confocal images validating the drawings must be provided.

We thank the reviewer for this suggestion and include the original immunofluorescence images (new Fig. 4a, pg. 13)

- Fig.6 and Fig.7: I miss a deeper characterisation of the mt-Ca²⁺ sensor here and the specificity of mitochondrial- and myelin-related interventions on the mt-Ca²⁺ signal. Already, it would be easier to appreciate Fig.6b, if time lapse images of given mitochondria would be provided and the dynamic range of mt-Ca²⁺ signal over time would be depicted.

This is an important point and we have included more time lapse images of the mt-GFP responses (revised Fig. 7b,e, pg. 21). We have also added Supplemental Videos S1 and S2, showing calcium responses in mitochondria and the cytosol, respectively.

In functional terms, how does AP-induced mt-Ca²⁺ signal propagates in nearby mitochondria, would this be impaired in genetic models of myelin loss, by connexin blockers or in Cx47 or Cx32 KO mice? Would interfering with mitochondrial function directly show up in the mt-Ca²⁺ signal and how would this be affected in control mice or after demyelination?

Regarding a 'deeper characterization' and specificity of the demyelination experiments we have added in the revision new calcium imaging of the interneuron axonal cytoplasm (AAV GCaMP6s injected in PV-Cre mice). These results reveal highly variable [mt-Ca²⁺]_i as a function of myelination. The lack of mt-Ca²⁺ in PV internodes is most likely downstream from the electrical insulating features of myelin to limit axolemmal depolarization. Based on these and other data we assume that clustering of mitochondria to myelinated segments of PV internodes may be achieved via oligodendrocyte-specific cues delivered via the noncompacted myelin cytoplasm (Figure 6, pg. 18) rather than axolemmal Ca²⁺ influx. Studying the connexin knockout mice exceeds the scope of this study.

Reviewer #2 (Remarks to the Author):

Axons of parvalbumin +ve inhibitory interneurons in the cerebral cortex are extensive and have a complex arborisation. They are partially myelinated and the patterning of myelin is influenced by the shape of the axonal arborisation, such that sheaths tend to be more frequent proximally and are absent at branch points and en passant boutons. The function of myelin on Pv +ve interneurons is not fully understood.// . The work is original and provides new information in relation to both aspects. The results are compatible with the conclusion that “myelin [or functional consequences of myelination] clusters mitochondria and diversifies mitochondrial Ca²⁺ buffering” on Pv +ve interneurons. It is interesting to speculate that the findings “indicate[s] that myelination at the level of branch orders may provide an axoglial mechanism to regulate energy homeostasis in PV+ axons”, but this.

We thank the reviewer for their in-depth analysis and are pleased by their positive comments on the originality and novelty of our study. The sentence ended abruptly with “ ... but this” so we are unsure whether this was an action point. Nonetheless, we hope the revised version and new data described below addresses most of the concerns of the reviewer.

The analyses and methodologie appears sound // As the results are in contrast to previous observations in white matter and/or glutamatergic neurons, where mitochondrial volume is increased in non-myelinated regions of otherwise myelinated axons (e.g. in the optic nerve) it would seem appropriate for the authors to show they can replicate previous work, using the method adopted here. They may already have data from L5 PNs or could incubate ex vivo optic nerve, including the non-myelinated nerve head, with AAV.

The reviewer correctly indicates our findings in PV+ interneurons contrasts with previous studies in which demyelination typically increases the mitochondrial content. In the revised version we have addressed this point at several levels. First, new analyses from 3D EM blocks compares the patchy myelinated of L2/3 pyramidal neuron axons and basket cell interneuron axons and shows that PV interneuron mitochondria are differentially organized by myelination (see new Fig. 5, pg. 16). Secondly, we have now added data from Rbp4-Cre axons in the white-matter (new Supplementary Fig. 5, Supplementary Information pg. 7). Consistent with Ohno et al. (PNAS, 2014) we find that cuprizone-induced demyelination significantly increases the number of mitochondria. These data suggest that PV interneuron axons are indeed differently affected by myelin loss.

If feasible, it would be interesting to know whether increased mitochondrial clustering is dependent upon compact myelin being present, or whether ensheathment per se is sufficient. This should be possible in shiverer mice and would help determine whether insulation (a function of compact myelin) is required for mitochondrial clustering.

We fully agree such experiments could be helpful (see also request of ‘complementary models’ by Reviewer #1). In the revised version we present new experimental data based on a triple transgenic mouse (PV-Cre; Ai14; Shiverer) and injected with mt-GFP AAV (Fig. 6, pg. 18). Following the patch-clamp recordings we made immunofluorescent staining and used myelin oligodendrocyte glycoprotein (MOG) to identify axonal branches which are in contact with the oligodendroglial membranes and those branches which lack such glia wrapping. Interestingly, consistent with the role of cytoplasmic spaces to metabolically fuel axons (Saab, Nave, (2017) *Curr Opin Neurobiol* 47, 104–112; Lee et al. (2012) *Nature* 487, 443–448; Fünfschilling, et al. (2012) *Nature* 485, 517–521) the results show that compact myelination is not required for mitochondria clustering and size increase. Oligodendroglial wrapping with noncompacted membranes (identified by MOG) suffices to cluster mitochondria to the PV

internodes. Interestingly, consistent with Andrews et al. 2006 we see increased number of mitochondria in the Shiverer but only in *unmyelinated* axonal segments (Fig. 6h, pg. 18). The new results further support the notion that axon-glia signaling via inner oligodendrocyte cytoplasmic loops suffices to stabilize mitochondria in PV internodes and ‘fuel’ the axon.

Minor points:

Perhaps the authors could speculate a little more on how their observations may shed light on the susceptibility of PV+ interneurons in multiple sclerosis.

We agree that the implications of our finding of a direct coupling between mitochondria and oligodendroglial wrapping suggest that PV interneurons may be more vulnerable to myelin loss. In the revised Discussion we included some ideas (pg. 28). However, more research is required to examine whether the myelin loss acutely affects the energy supply in interneurons, and how this differs from excitatory axons.

Intro Lines 62-64 Consider citing PMID: 16555298, which describes the distribution of mitochondria in axons that are never surrounded by compact myelin, in shiverer mice, which have otherwise healthy axons, and 14736793, which examines mitochondrial distribution in myelinated and non-myelinated regions of the healthy adult human optic nerve.

We thank the reviewer for these references. The human optic nerve study is somewhat distant from the present topic, but we have included the Andrews et al article (*J Neurosci Res* 83:1533–1539, 2006) which is indeed relevant to the interpretation of our results.

“being partially electrically separated from the myelin sheath” meaning what?

We rewrote the discussion, and this sentence was removed.

Result Line 07. It might be clearer to mention the specific Cre driver lines at this point.

We rewrote this in the revised version of the manuscript (pg. 4).

Results Lines 66-67. What is the evidence that 5 mM L-lactate and 10 mM glucose resembles the in vivo cellular environment?

We provide a reference for and clarification of this claim (pg. 7).

Page 14, lines 34-35. Suggest changing “Myelin loss is typically associated with” to “Myelin loss can lead to”

We thank the reviewer for the suggestion. In the revision, this sentence has however been removed.

Is the graph in 4K a cumulative distribution plot?

Yes it is. This was incorrectly not mentioned in the legend but is now better clarified in the revision (see legend Fig. 3, pg. 12).

Page 17, “*These data suggest that in PV+ axons, branch points are sites of nodes of Ranvier and mitochondria are clustered to the nodal domains.*” Consider citing PMID 18381760, which shows that in small diameter myelinated axons in white matter, there appears to be no clustering of mitochondria at paranodal/nodal regions of small diameter fibres.

We have cited this article in the results section describing the 3D EM data, where we find that mitochondria in PV⁺ axons also avoid the paranodes (pg. 14).

Figure 7C, does this refer to a specific population of mitochondria only?

No, these are all mitochondria distances relative to Caspr.

Discussion, line 71 “...but also impedes transport and metabolic supply” seems not entirely clear.

In the Revision we have reworded this part of the Discussion and now included citations for these ideas (pg. 26).

Reviewer #3 (Remarks to the Author):

In this paper, Kole et al. study, how in parvalbumin-positive (PV) interneurons myelination affects mitochondrial morphology, distribution and function – under normal conditions and during cuprizone-induced demyelination. // ... overall I appreciate the chosen experimental model, the combination of techniques that integrates electrophysiology in the characterization of the neurons (which is missing in many other comparable papers) and the care with which the data are reported, quantified and illustrated. Also, the results suggest that PV interneurons might indeed differ in some aspects of their mitochondrial distribution and demyelination response from the excitatory neurons analyzed so far.

We thank the reviewer for recognizing the technical approaches. We have substantially added new data (e.g. new Fig. 5, pg. 16) to strengthen our finding the PV interneuron axons differ from excitatory ones.

Still, the paper left me unsatisfied with the level of new mechanistic insight actually gained. A substantial part of the paper shows controls or confirms known facts (Fig 1-3 do not add too much to the core of the story and could actually be supplementary: Fig 1 shows technical matters that are not too surprising, Fig 2 data that do not really relate to mitochondria in the end and Fig 3 shows facts established for many neurons already).

In the revised version we have moved the original Figure 3 to the supplements (Supplementary Information pg. 4). Furthermore, we have added additional new experimental data supporting our main conclusion.

Major points:

1) A weakness is in my view the limited (experimental) analysis of the calcium imaging data. The authors write about ‘calcium buffering’, which has an interpretative quality, suggesting that mitochondrial calcium uptake here is serving to stabilize (“buffer”) calcium levels elsewhere – in the discussion it then seems that the authors are more concerned with the metabolic regulation that such different calcium levels or peaks could imply (which is rather linked to the calcium-dependence of some key metabolic enzymes, not a possible “buffering” function). / .../ But it remains unclear to me why the authors undertake no experimental effort to explore these different possibilities, e.g. by parallel measurements of cytoplasmic calcium levels around the respective mitochondria populations (which should be technically feasible), but perhaps also estimates of mitochondrial potential (which might be more difficult, but at least such measurements have been reported in some in vivo systems – and might be more precise in a systems where they can be internally normalized by comparing mitochondria in different parts of the same axon).

We agree with the reviewer that distinguishing between cytoplasmic $[Ca^{2+}]_i$ and mitochondrial $[Ca^{2+}]_m$ is critical to interpret the mt-GCaMP responses. In the revised manuscript we have now added a substantial amount of new experimental data. First, we have used cytoplasmic GCaMP (AAV1-CAG-Flex-mRuby2-GSG-P2A-GCaMP6f) injected in PV-Cre; Ai14 mice) to study the cytosolic Ca^{2+} responses. While we have not measured this calcium simultaneously with mt-GCaMP6f, these results show, as predicted, that myelinated PV branches (MBP positive) show little, if any, calcium influx compared to parent or daughter branches in the same axon arbor which are unmyelinated (MBP negative) (new Fig. 7e-f, pg. 21). Measuring mitochondrial membrane potential is an important avenue for future studies.

2) The authors make the intriguing observation that the proximal axon segments gain length during cuprizone-induced demyelination. This is surprising, as it seems hard to imagine how this could work simply given the density of cortical neuropil. /.../ Overall it seems that in Fig. 4 the total number of mitochondria per PV axon is the most stable value (although it is not directly reported). While of course for the measure of mitochondrial density, changes in nominator or denominator have essentially the same effect, cell biologically and mechanistically it makes a big difference, whether axons locally expand in length or mitochondria are locally added or removed. So these options should be experimentally distinguished.

We agree these data remained ambiguous about the contribution of mitochondria and segment expansion. In the revised version we now provide substantial new data in support of a mitochondria-specific change instead of axon cytoplasm changes. First, examining in more detail the MBP⁺ segments we now show directly that the number of mitochondria relative to the segment are different (new Figure 4, pg. 13). Secondly, at much greater spatial resolution in 3D EM we find independent evidence that mitochondria are not only changed in number but also in volume (Figure 5, pg. 16). Although these data do not explain why we find length changes in the proximal branches following cuprizone, they are supportive of a direct role of myelin. Thirdly, based on new experiments with the *shiverer* mice we conclude that noncompacted oligodendrocyte cytoplasm is essential for the mitochondria assembly at PV interneuron segments (Figure 6, pg. 18).

3) The observation of increased mitochondrial density under myelinated vs. non-myelinated segments is indeed notable and counterintuitive / / However, it has been well appreciated that within a myelinated axon segment the mitochondrial distribution often is not uniform (e.g. with low density in the center of internodes and the nodes themselves, but relative accumulations in the para/juxta-nodal areas). So it would be informative to know how mitochondria are distributed within the myelinated and non-myelinated areas, and whether the density changes systematically at a border area ...

Using the 3D EM data we have now plotted the distribution relative to the edge of the myelin (new Supplementary Fig. 6c, Supplementary Information pg. 8). We find that the mitochondria are mostly found within the internode, less near paranodal loops. We find mitochondria in ~60% of branch points.

After all, it remains somewhat unclear how local the effects of a mitochondrion as an ATP source are, and how far e.g. ATP, glucose, lactate or other small metabolites could diffuse inside the axoplasm. In the discussion, the authors seem to argue that perhaps the prevailing assumptions about energy demand in internodes are not universally true – and based on the difference between this report on PV axons and the axon populations previously analyzed (e.g. in the optic nerve) the suggestion is that different axon classes have different local energy demands. The authors e.g. point to axon diameter and surface/volume ratios and sub-myelin ion transients. Can this be analyzed, e.g. by looking for additional local correlations in axon diameter and mitochondrial density? Would it be possible to locally measure ATP levels and dynamics using a biosensor such as ATEAM (as done by the Nave lab in optic nerve axons)?

We thank the reviewer for these suggestions. Using the 3D EM data, in the revised manuscript we have now added analyses on mitochondrial volume with respect to the surrounding cytoplasmic volume (new Fig. 5, pg. 16). These new data showed that basket cells have larger mitochondria, also when correcting

for the local cytosolic volume of myelinated segments, there is significantly more mitochondria in PV⁺ internodes compared to pyramidal neuron axons.

Regarding the local measurement of ATP levels, this is a very good suggestion. However, we believe that these experiments are beyond the scope of the current study as they are likely not trivial and will require significant fine-tuning. For example, an important consideration is the dynamic range of the sensor, which may not be sufficient to detect the (likely small) differences in two neighbouring axonal branches. We do however mention in the discussion (pg. 26, 27-28) that such experiments will benefit our understanding of the ATP dynamics in myelinated, unmyelinated and demyelinated PV⁺ axons.

4) The authors openly discuss the shortcoming of their analysis using conventional light microscopy, which is laudable. At the same time, it seems to me that given the availability of annotated large scale 3D EM data sets, it should be possible to confirm some results in Fig. 5 experimentally at ultrastructural resolution.

We fully agree with the reviewer these data sets can provide highly valuable insights. We have analyzed the “Cortical mm³” data set from MICrONS (<https://www.microns-explorer.org/cortical-mm3>). In the revised version of the manuscript, we describe the independent evidence for increased mitochondrial content in myelinated branches from basket cell interneurons (new Fig. 5, pg. 16). Secondly, we find a lack of clustering along the heterogeneously myelinated layer 2/3 pyramidal neuron axons. In addition, the data are also quantitatively in line with our immunofluorescence data (new Fig. 4, pg. 13). We believe these new results provide exciting support for an interneuron-specific mitochondria distribution by myelination.

Minor comments:

Fig 1c vs f: It would be more comparable to show in both cases (PV and Rbp4) the Ai-labeled population and/or a filled single neuron – ideally both in three channels.

We updated this figure to show new high-resolution images demonstrating the co-localization of tdTomato and mt-GFP (new Fig. 1c, e, pg. 6).

Fig. 1d not really necessary

We agree and have removed this from Fig. 1. We now report the specificity in the main text (pg. 4).

Reviewers' Comments:

Reviewer #1:

Remarks to the Author:

No further questions.

Reviewer #2:

Remarks to the Author:

The revised manuscript provides compelling evidence that in PV+ axons, mitochondrial function and spatial distribution are influenced by ensheathment of axons by oligodendroglial cell membrane. This might be related to the transfer of metabolites from the glial cell to the axon, along the internode, although at this stage, that remains speculative. The observations are novel; they suggest that amongst several functions of the oligodendrocyte/myelin, there are neuronal-type specific ones; they might eventually shed light on the reasons why this neuronal population is particularly vulnerable to injury in MS.

Minor comments:

Lines 612-614 As far as I know, CNPase has never been shown to be involved in lactate and pyruvate supply, although, given its role in keeping the cytoplasmic channels of myelin open, it can be speculated that it facilitates the access of metabolites to the glial-axonal junction.

In relation to shiverer mice, it is probably more accurate to refer to 'ensheathment' than to 'myelin' e.g. "Myelinated segments of PV+ axons were also shorter..."

Reviewer #3:

Remarks to the Author:

I thank the authors for their constructive response to the reviews. They have addressed all my comments - and I agree that some of my suggestions exceeded the scope of the current work, especially if the same high technical standard is to be maintained. Overall, I feel the manuscript now provides a clear picture that there is substantial diversity of how different neurons organize their mitochondria with regard to their myelination pattern - which is a new and (for me) surprising insight. I think the paper now also provides one of the most thorough descriptions of the distribution of mitochondria in two classes of CNS neurons, which will give it substantial impact. Thus, I strongly support publication of the revised manuscript, and congratulate the authors on this impressive and comprehensive study.

Only a comment to Fig.1 - it seems to contain a swapped panel in e. While I would have preferred to also see global labeling of the Rbp cells rather than a fill (and perhaps fills of both cell types in addition), I understand the authors' choice - and would just suggest double-checking that nothing else has been confused in the revision process.

We thank the reviewers for their comments on our revised manuscript and are pleased to learn that the additional data has adequately addressed their previous points. Below we provide point-by-point responses to their comments on the revised manuscript.

Reviewer #1 (Remarks to the Author):

No further questions.

Reviewer #2 (Remarks to the Author):

The revised manuscript provides compelling evidence that in PV+ axons, mitochondrial function and spatial distribution are influenced by ensheathment of axons by oligodendroglial cell membrane. This might be related to the transfer of metabolites from the glial cell to the axon, along the internode, although at this stage, that remains speculative. The observations are novel; they suggest that amongst several functions of the oligodendrocyte/myelin, there are neuronal-type specific ones; they might eventually shed light on the reasons why this neuronal population is particularly vulnerable to injury in MS.

Minor comments:

Lines 612-614 As far as I know, CNPase has never been shown to be involved in lactate and pyruvate supply, although, given its role in keeping the cytoplasmic channels of myelin open, it can be speculated that it facilitates the access of metabolites to the glial-axonal junction.

We thank the reviewer for their positive reaction to the additional data and helpful comments. We agree that it remains to be experimentally demonstrated that the cytoplasmic channels help with the transfer of pyruvate and lactate. We have adjusted this in the discussion and wrote "...which may be involved in lactate and pyruvate supply to the periaxonal space"

In relation to shiverer mice, it is probably more accurate to refer to 'ensheathment' than to 'myelin' e.g. "Myelinated segments of PV+ axons were also shorter..."

We agree that in the case of *Shiverer* mice, it is more appropriate to indicate that the ensheathed axonal segments are not equal to myelination. We have therefore more clearly written these axons possess "*noncompacted myelin*".

Reviewer #3 (Remarks to the Author):

I thank the authors for their constructive response to the reviews. They have addressed all my comments - and I agree that some of my suggestions exceeded the scope of the current work, especially if the same high technical standard is to be maintained. Overall, I feel the manuscript now provides a clear picture that there is substantial diversity of how different neurons organize their mitochondria with regard to their myelination pattern - which is a new and (for me) surprising insight. I think the paper now also provides one of the most thorough descriptions of the distribution of mitochondria in two classes of CNS neurons, which will give it substantial impact. Thus, I strongly support publication of the revised manuscript, and congratulate the authors on this impressive and comprehensive study.

Only a comment to Fig.1 - it seems to contain a swapped panel in e. While I would have preferred to also see global labeling of the Rbp cells rather than a fill (and perhaps fills of both cell types in addition), I understand the authors' choice - and would just suggest double-checking that nothing else has been confused in the revision process.

We are thankful for the reviewer's positive evaluation of the revised manuscript and constructive comment. We have corrected the mistake in Figure 1e panel.